# Recalibrating timing behavior via expected covariance between temporal cues

Benjamin J De Corte[1], Rebecca R Della Valle[2], Matthew S Matell[3]*

[1]Iowa Neuroscience Institute, The University of Iowa, Iowa City, United states; [2]Department of Psychological and Brain Sciences, The University of Delaware, Newark, United states; [3]Department of Psychological and Brain Sciences, Villanova University, Villanova, United states

**Abstract** Individuals must predict future events to proactively guide their behavior. Predicting when events will occur is a critical component of these expectations. Temporal expectations are often generated based on individual cue-duration relationships. However, the durations associated with different environmental cues will often co-vary due to a common cause. We show that timing behavior may be calibrated based on this expected covariance, which we refer to as the 'common cause hypothesis'. In five experiments using rats, we found that when the duration associated with one temporal cue changes, timed-responding to other cues shift in the same direction. Furthermore, training subjects that expecting covariance is not appropriate in a given situation blocks this effect. Finally, we confirmed that this transfer is context-dependent. These results reveal a novel principle that modulates timing behavior, which we predict will apply across a variety of magnitude-expectations.
DOI: https://doi.org/10.7554/eLife.38790.001

*For correspondence:
matthew.matell@villanova.edu

**Competing interests:** The authors declare that no competing interests exist.

## Introduction

Being able to predict future events provides observers with a substantial adaptive advantage when navigating the environment. Individuals can use these predictions to proactively guide their behavior and, without them, would be forced to passively react to stimuli as they occur in the present moment.

Anticipating *when* future events will occur is a central component of these predictions and is an important area of research across fields such as psychology and neuroscience, spanning topics including attentional orienting (*Beck et al., 2014*; *Coull et al., 2016*; *Nobre and van Ede, 2018*), decision-making (*Akdoğan and Balcı, 2017*; *Jazayeri and Shadlen, 2010*; *Luzardo et al., 2017*; *Simen et al., 2016*), reinforcement learning (*Lau et al., 2017*; *Rivest et al., 2014*; *Sutton, 1988*), memory storage (*Gallistel, 2017*; *Johansson et al., 2015*), and treatments for cognitive impairments in disease (*Emmons et al., 2017*; *Gu et al., 2015*). Temporal expectations can be guided by local sources of information. For example, specific environmental stimuli often function as temporal cues (e.g., a traffic light turning yellow indicates that it will turn red after a certain duration has passed).

However, global properties of the environment can be used to calibrate these expectancies. This point can be illustrated with the following metaphor. Consider an animal that is foraging for food in an area that contains two plant types—plant A and plant B. When it searches through plant A, it typically finds food every 8 s. In contrast, when searching through plant B, it finds food every 16 s. Despite the fact that these plants (i.e. temporal cues) are associated with different inter-food intervals (i.e. durations), the intervals for both plant types will co-vary as environmental conditions change. For example, during a drought, both plant types would produce less food than normal.

Consequently, the intervals would increase for both plant types (i.e. food would be found less frequently), relative to stable weather conditions.

Now, consider what would happen if the animal were to enter the drought-impacted area and learned that plant A's interval increased. If the animal anticipates that there will be covariance between plant A and B's intervals, it could use plant A's new inter-food interval to estimate what plant B's new duration will be, without physically visiting the plant itself. In other words, knowledge of or expectations about the statistical structure of the environment would allow the learning from plant A to generalize to other plants in the area.

From an evolutionary standpoint, this would be a highly advantageous process because the animal could predict the expected rate of return for foraging in the entire area based on a change in one interval. As such, it would spend less time before deciding to stay or leave a given location, relative to learning each plant's new interval individually.

The essential point proposed in the above example is that, in the environment, the intervals associated with different temporal cues will often co-vary due to a common underlying cause (e.g. drought, increased rainfall, seasonal changes, etc.). Our question is whether a mechanism has evolved that capitalizes upon this property in order to adaptively recalibrate behavior. We will refer to this possibility as the 'common cause hypothesis' and test its predictions in the following experiments.

## Results

### Experiment 1: changes in temporal expectations transfer across cues

The underlying premise of the common cause hypothesis is that common causal factors can often produce covariance among the durations associated with distinct temporal cues. According to this hypothesis, observers can benefit from this statistical property by anticipating that, when the duration associated with one temporal cue changes, the durations associated with other cues will have changed in the same direction.

We tested this hypothesis using the 'peak-interval' procedure. In this task, trials begin with the presentation of a cue that indicates reward can be earned for responding after a specified duration elapses (e.g. responding at a nosepoke after 16 s have passed). Probe trials are also included in which the cue remains on for much longer than normal (e.g. 48 –64 s) and no reward is provided. When averaged across trials, probe trial responding resembles a Gaussian curve with a peak (i.e. peak time) centered over the trained time of reward, providing an estimate of the temporal expectation associated with the cue in question (for examples see *Figure 1*).

Using this task, we trained rats to associate two cues (tone and light) with either an 8 or 16 s delay to reward availability (e.g. tone-8 seconds/light-16 seconds; counterbalanced). Then, during a 'change phase', we altered the duration of one of the cues, hereafter, referred to as the 'changed cue'. Specifically, for one group, we decreased the short, 8 s cue's duration to 4 s (8-to-4 group). Conversely, for the other group, we increased the long, 16 s cue's duration to 32 s (16-to-32 group). Importantly, we did not present the other, 'unchanged cue' until testing, which occurred after rats had adapted to the changed cue's new duration (see *Figure 1A,D* for design schematics).

According to the common cause hypothesis, increasing or decreasing the changed cue's duration should produce a corresponding increase or decrease in the unchanged cue's temporal expectation. We evaluated this during a test phase by introducing probe trials for the unchanged cue (i.e. no feedback regarding when to respond was provided). Normalized average probe trial responding from the 8-to-4 and 16-to-32 groups during initial training and testing are plotted in *Figure 1B* (see *Figure 1—figure supplement 1* for non-normalized graphs). For clarity, we present percent change in peak-times in *Figure 1C and D*. However, all statistical comparisons were conducted on non-normalized values here and throughout, with the exception of the Bayesian analyses to account for within-subject variance (see below and methods). Consistent with our prediction, in both groups, responding to the unchanged cue shifted in the same direction as the changed cue, despite the fact that the unchanged cue's duration was never explicitly altered.

Specifically, in the 8-to-4 group, decreasing the short cue's duration from 8 to 4 s caused responding to the long (i.e., unchanged) cue to shift leftward [$M = -22\% \pm 5\%$ *SEM*, $t(9) = -4.10$, p<.005; ANOVA: Phase, $F(1,8) = 55.72$, p<0.001; Cue, $F(1,8) = 875.72$, p<0.001]. Likewise, in the 16-

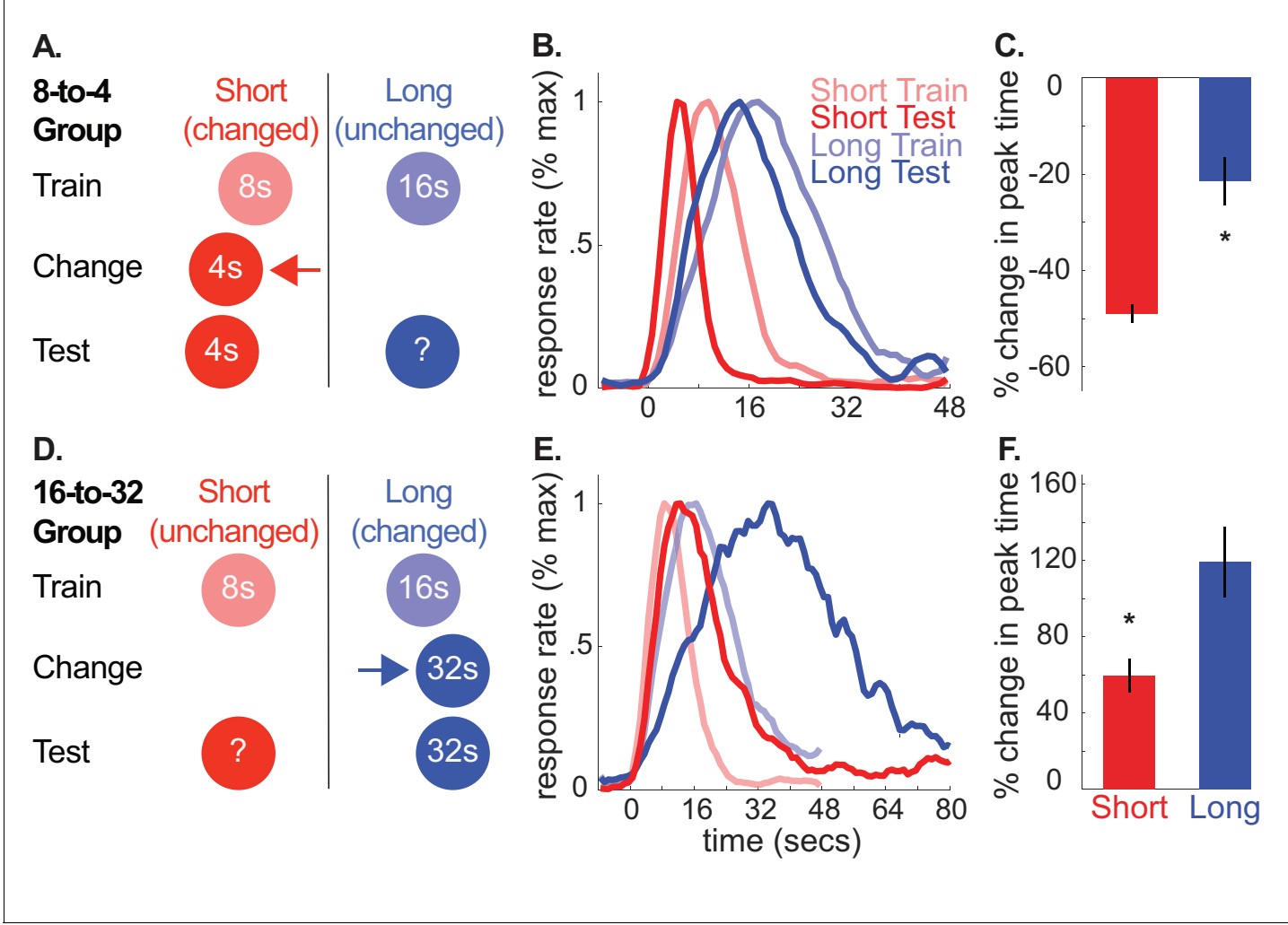

**Figure 1.** A and D depict the general design used for the 8-to-4 (*n* = 10) and 16-to-32 (*n* = 10) groups, respectively. (B) and (E) show average normalized response rate during probe trials during initial training and testing for the 8-to-4 and 16-to-32 groups, respectively. For presentation, each curve was normalized by the maximum value and smoothed over a 6-bin window. (C) and (F) show percent-change in peak time during testing (±SEM), relative to initial training for the 8-to-4 and 16-to-32 groups, respectively. Stars indicate significance (p<0.05).

DOI: https://doi.org/10.7554/eLife.38790.002

The following source data and figure supplements are available for figure 1:

**Source data 1.** Data used to generate *Figure 1*.

DOI: https://doi.org/10.7554/eLife.38790.010

**Source data 2.** Table showing percent change in peak, start, and stop times (+/- SEM) for the unchanged cue across all experiments.

DOI: https://doi.org/10.7554/eLife.38790.009

**Figure supplement 1.** Raw probe trial response rates during training and testing for all experiments.

DOI: https://doi.org/10.7554/eLife.38790.003

**Figure supplement 1—source data 1.** Data used to generate *Figure 1—figure supplement 1*.

DOI: https://doi.org/10.7554/eLife.38790.004

**Figure supplement 2.** A and B depict design schematics (left panels) and average percent change in start/stop times (±SEM; right panels) for the 8-to-4 and 16-to-32 groups for Experiment 1, respectively.

DOI: https://doi.org/10.7554/eLife.38790.005

**Figure supplement 2—source data 1.** Data used to generate *Figure 1—figure supplement 2*.

DOI: https://doi.org/10.7554/eLife.38790.006

**Figure supplement 3.** Same as *Figure 1—figure supplement 2*, but showing single-trial averages (±SEM) for the remainder of the groups in the experiments (A, B, C, D, and E correspond to the 8-to-12 group of Experiment 2, both groups included in Experiment 3, the two Experiment 4 groups, and Experiment 5, respectively).

*Figure 1 continued*

DOI: https://doi.org/10.7554/eLife.38790.007

**Figure supplement 3—source data 1.** Data used to generate *Figure 1—figure supplement 3*.

DOI: https://doi.org/10.7554/eLife.38790.008

to-32 group, increasing the 16 s cue's duration to 32 s caused responding to the short (i.e. unchanged) cue to shift rightward [$M$ = 59% + /- 9% *SEM*, $t(9)$ = 6.02, p <. 001; ANOVA: Phase, $F(1,8)$ = 49.47, p<0.001; Cue, $F(1,8)$ = 363.02, p<0.001; Phase X Cue, $F(1,8)$ = 68.40, p<0.001]. In contrast, no significant changes were found in the relative spreads of the response function (i.e. coefficient of variation (CV) = spread/peak time) [8-to-4, $M$ = −4% ± 10% *SEM*, $t(9)$ =.038; p>0.05; 16-to-32, $M$ = 16% + /- 7% *SEM*, $t(9)$ = 2.21, p>0.05]. In other words, changes in the spread of responding were consistent with the scalar property of interval timing, whereby the spread scales in direct proportion to the peak time (*Gibbon et al., 1984*).

It is important to note that the smooth response curves shown in *Figure 1* do not reflect the response pattern generated on individual probe trials (*Church et al., 1994*). Rather, during probe trials, rats emit an abrupt burst of responses that typically start before and stop after the expected time of reward. While single-trial behavioral data during unchanged cue trials were limited and variable—a common finding during extinction trials in this task—changes in start and stop times were largely consistent with the peak time shifts (see *Figure 1—figure supplement 2* for single-trial averages for this experiment, and *Figure 1—figure supplement 3* for all others). Specifically, in the 16-to-32 group, both start and stop times were delayed during unchanged cue trials, relative to initial training [Start: $M$ = 59% + /- 12% *SEM*, $t(9)$ = 4.14, p<.005; Stop: $M$ = 65% + /- 10% *SEM*, $t(9)$ = 6.23, p<0.001]. Comparably, in the 8-to-4 group, long cue stop times were leftward shifted [$M$ = −27% ± 5% *SEM*, $t(9)$ = −5.54, p<0.001]. However, start times were not reliably different [$M$ = −4% ± 12% *SEM*, $t(9)$ = −0.83, p>0.05].

We also ran a control group, in which we replicated the design used for the 8-to-4 group, yet kept the short cue's duration at 8 s throughout the experiment (see *Figure 2* for schematic). This allowed us to assess the possibility that over-exposing rats to the changed cue during what would

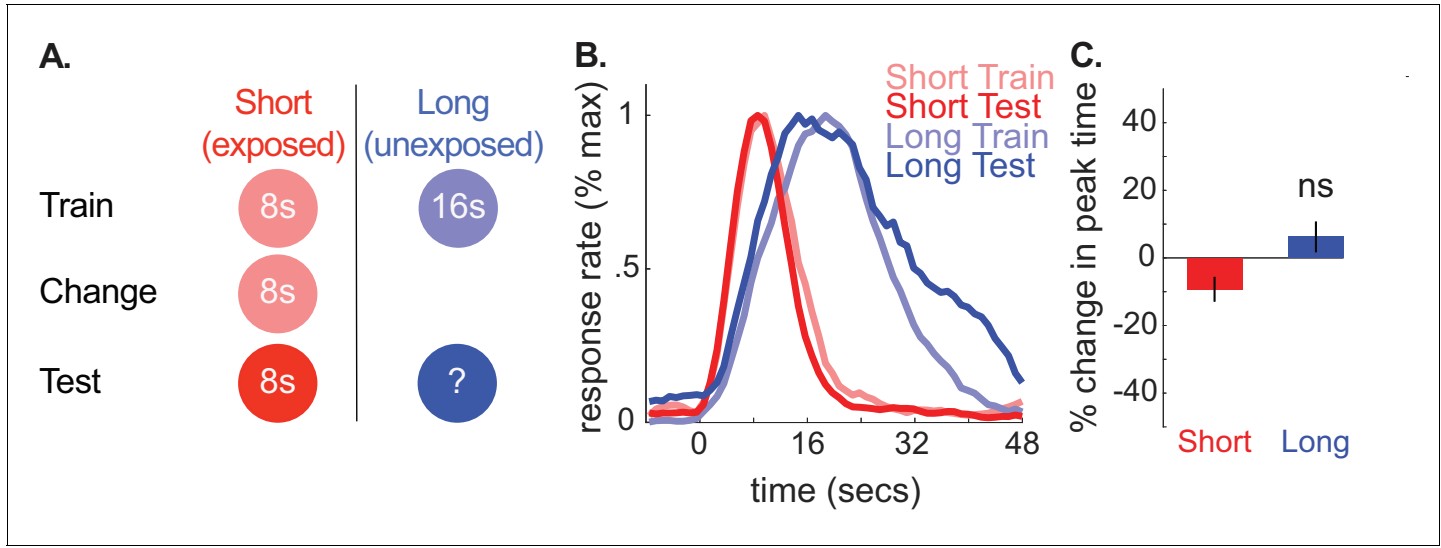

**Figure 2.** A depicts the design used for each phase in the no-change control group ($n$ = 10). (**B**) shows normalized response rate during training and testing. (**C**) shows percent change in peak times during testing (±SEM), relative to initial training.

DOI: https://doi.org/10.7554/eLife.38790.011

The following source data is available for figure 2:

**Source data 1.** Data used to generate *Figure 2*.

DOI: https://doi.org/10.7554/eLife.38790.012

have been the 'change phase' could cause responding to the long cue to shift leftward, regardless of whether a duration-change occurred.

Consistent with our hypothesis, we did not find reliable transfer to the long (unexposed) cue in this group. Specifically, peak times did not change reliably and, if anything, shifted slightly rightward [$M$ = 6% + /- 5% $SEM$, $t(9)$ =.86, p>0.05]. Furthermore, long-cue shifts were larger in the 8-to-4 group than the control group [ANOVA: Phase X Group, $F(1,16)$ = 16.24, p<.005]. Correspondingly, no CV changes were observed for the unchanged cue during testing [$M$ = 4% + /- 8% $SEM$, $t(9)$ =.46, p<0.05]. We also ran two Bayesian analyses (see methods) to assess evidence in favor of the null hypothesis that peak times did not shift during testing in the control group. The first used an uninformative prior (i.e. assessing evidence for a non-specific peak time shift). As expected, this showed substantial evidence in favor of the null [Bayes Factor = 3.34]. Critically, using an alternative prior that tested the specific hypothesis that the peak times shifted to the same degree as the 8-to-4 group showed decisive evidence in favor of the null [Bayes Factor = 2875.88]. Finally, neither start nor stop times reliably shifted in the no-change group [Start, $M$ = 17% + /- 11% $SEM$, $t(9)$ = 1.55, p>0.05; Stop, $M$ = 3% + /- 5% $SEM$, $t(9)$ =.61, p>0.05].

Together, these results provide support for the common cause hypothesis. When the duration associated with one cue changed, responding to the other cue changed in the same direction, as if rats expected that a common cause shifted the durations associated with both cues in a correlated manner.

## Experiment 2: cross-cue transfer does not result from regression to the mean

We propose that, in Experiment 1, the transfer to the unchanged cue occurred because subjects expected the durations associated with both cues would co-vary due to a common cause. For example, in the 8-to-4 group, responses to the long, unchanged cue shifted leftward because the short cue's duration shifted leftward.

However, this explanation highlights that altering the changed cue's duration will induce uncertainty regarding the status of the unchanged cue's duration. Many have shown that, when subjects time multiple durations under conditions of uncertainty, their estimates regress toward the mean of all experienced intervals (*De Corte and Matell, 2016b*; *Gu et al., 2015*; *Jazayeri and Shadlen, 2010*; *Shi et al., 2013*). This is often interpreted as a Bayesian process, wherein heightened uncertainty causes subjects to rely more heavily on prior knowledge (all previously learned intervals) over current sensory measurements (timing of the present cue's duration). Consequently, estimates of all intervals regress toward one another.

This explanation can account for Experiment 1's results. For example, in the 8-to-4 group, the uncertainty from altering the 8 s cue's duration could have caused responses for the 16 s cue to regress leftward, toward the average of all previously experienced intervals (4, 8, and 16 s). Similarly, in the 16-to-32 group, increasing the 16 s cue's duration may have caused responses to the 8 s cue to move rightward, toward the mean of the previously learned intervals (8, 16, and 32 s).

This 'regression to the mean' account and the common cause hypothesis make identical predictions for Experiment 1. However, we were able to dissociate them with the following design (*Figure 3A*). First, we trained rats to associate a tone and light with either an 8- or 16 s duration, as in Experiment 1. Then, we increased the 8 s cue's duration to 12 s and subsequently evaluated how rats responded to the 16 s cue. In this case, the average of all intervals remained to the left of the 16 s cue's duration. Therefore, the regression account predicts a leftward shift in responding. In contrast, according to the common cause hypothesis, increasing the 8 s cue's duration to 12 s should cause responses to the 16 s cue to shift rightward, as the direction of the changed-cue shift should determine the direction of the unchanged-cue shift.

As depicted in *Figure 3B and C*, peak times for the unchanged, 16 s cue shifted rightward, consistent with the common cause hypothesis [$M$ = 13% + /- 5% $SEM$, $t(9)$ = 2.34, p<0.05]. As before, there were no reliable CV changes [$M$ = 13% + /- 7% $SEM$, $t(9)$ = 1.35, p>0.05]. Finally, there was a marginally significant rightward shift in start times and a reliable rightward shift in stop times [Start, $M$ = 18% + /- 10% $SEM$, $t(9)$ = 2.02, p=0.075; Stop, $M$ = 9% + /- 3% $SEM$, $t(9)$ = 2.91, p<0.05].

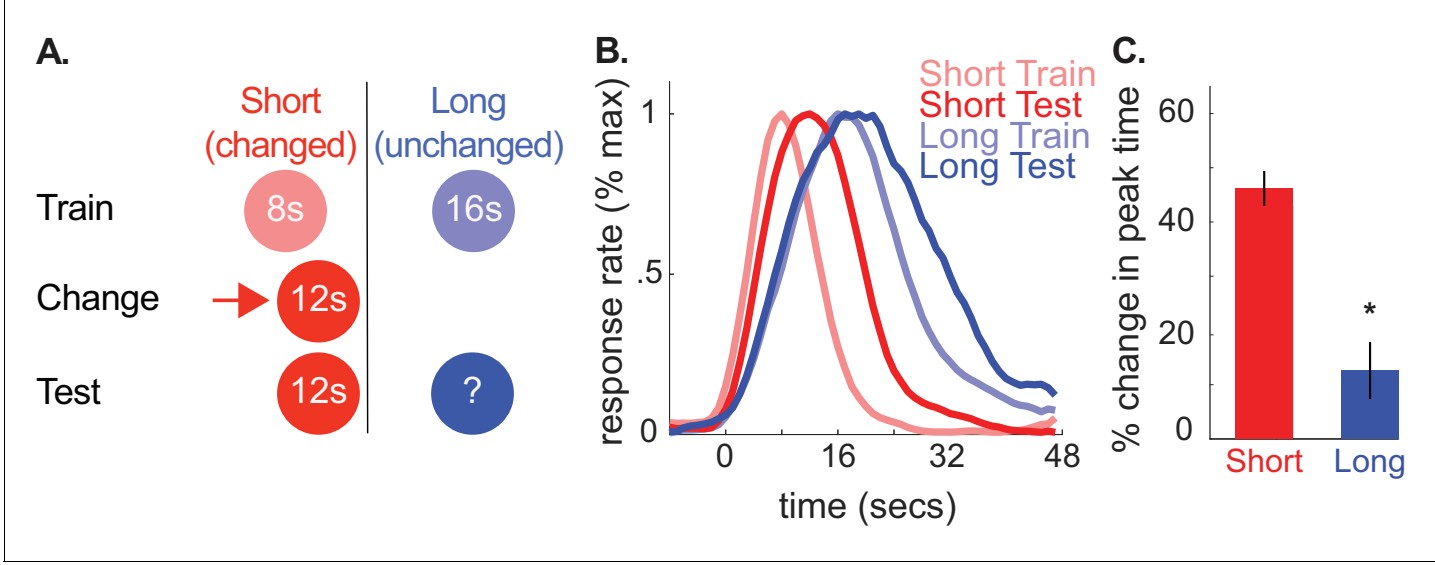

**Figure 3.** A shows the design used in the 8-to-12 group of the second experiment (*n* = 10). (B) shows normalized responding during training and the test phases. (C) shows percent change in peak time for each cue during testing (±SEM), relative to initial training. Stars indicate significance.

DOI: https://doi.org/10.7554/eLife.38790.013

The following source data is available for figure 3:

**Source data 1.** Data used to generate *Figure 3*.

DOI: https://doi.org/10.7554/eLife.38790.014

## Experiment 3: cue-duration change effects supersede reward structure change effects on transfer

In our previous experiments, altering the changed cue's duration constituted a shift in the local cue-duration relationship. However, it also altered the reward-structure of the overall task itself. Using the 8-to-4 group of Experiment 1 as an example, decreasing the 8 s cue's duration to 4 s caused a global decrease in the intervals being produced during the task. Furthermore, it introduced a novel duration that was earlier than the intervals the rats had learned up to that point. These non-cue-specific changes to the overall task could produce the pattern of transfer we observed.

For example, altering the changed-cue's duration often shifted the overall average time-to-reward during the task. For illustration, during the change phase in the 8-to-4 group, reward was delivered consistently at 4 s, rather than occurring at 8 or 16 s as during training. Unchanged-cue responding could have shifted in the direction of this average reward-time change. However, we note that, in the no-change group of the first experiment, maintaining the 8 s cue's duration during the 'change' phase decreased the overall average time-to-reward (i.e., consistently at 8 s, rather than 8 or 16 s during training). We did not observe a leftward shift in responding in this group, thereby weakening this explanation.

Another possibility is that introducing a novel duration that was earlier or later, relative to the previously trained intervals, caused responses to the unchanged cue to shift in the same direction. However, in the 8-to-12 group of Experiment 2, the new 12 s duration fell between the initial 8- and 16 s durations. As this new duration was neither earlier nor later than both of the two previously trained intervals, this explanation would not account for the rightward shift in unchanged-cue responding.

While these results weaken a reward-structure account, we directly addressed this concern in Experiment 3 (see *Figure 4A and B*). First, we trained rats to associate two cues with an 8- and 16 s duration, respectively. Then, we instituted a change phase, in which we stopped presenting the 16 s cue. In one 'no-change group', we kept presenting the 8 s cue, maintaining its duration. Critically, we also introduced a novel cue that predicted reward availability at 4 s. This changed the reward structure, without introducing a cue-duration relationship change. In contrast, in the 'change group', we decreased the 8 s cue's duration to 4 s and associated the novel cue with an 8 s duration. With

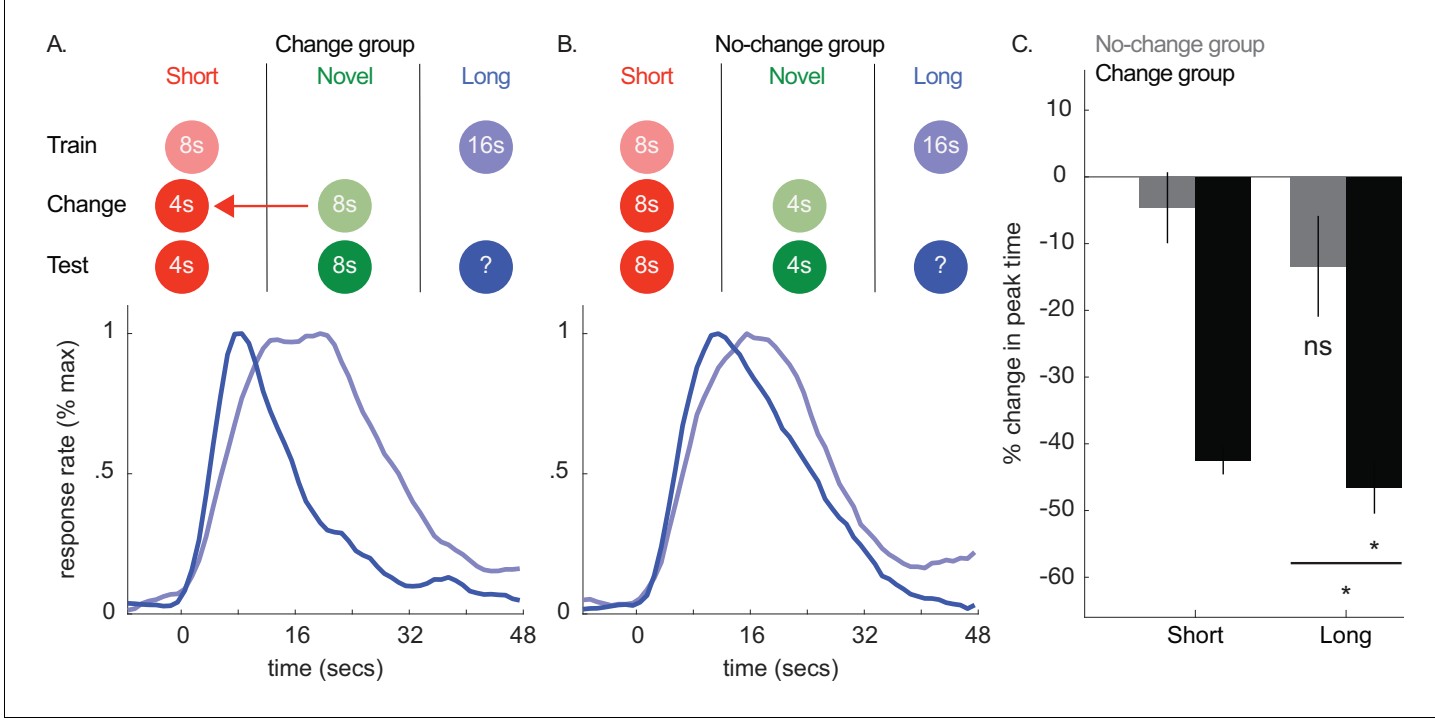

**Figure 4.** Top panels of A and B show designs used for the change (*n* = 5) and no-change (*n* = 5) groups of Experiment 3, respectively. Mean normalized response rate for the critical, long cue during initial training and testing are plotted in the corresponding bottom panels. (**C**) shows percent change in peak times for the short and long cues in each group during testing (±SEM).

DOI: https://doi.org/10.7554/eLife.38790.015
The following source data and figure supplements are available for figure 4:

**Source data 1.** Data used to generate *Figure 4*.
DOI: https://doi.org/10.7554/eLife.38790.018

**Figure supplement 1.** A and B show probe trial responding during initial training and testing for the short and long cues in the change (*n* = 5) and no-change (*n* = 5) groups of Experiment 3, respectively.
DOI: https://doi.org/10.7554/eLife.38790.016

**Figure supplement 1—source data 1.** Data used to generate *Figure 4—figure supplement 1*.
DOI: https://doi.org/10.7554/eLife.38790.017

this design, the overall reward structure of the task (i.e. cues, durations, etc.) was identical between the two groups. However, only the 'change' group experienced a cue-duration relationship change. Consequently, any between-group differences in unchanged-cue responding could only be driven by the cue-duration change.

Consistent with the common cause hypothesis, the magnitude of the long-cue peak time shift was far larger in the change-group than the no-change group [*Figure 4C*; Change group: *M* = −47% ± 4% *SEM*; *t*(8) = −7.89, p<0.05; No-change group: *M* = −13% ± 8% *SEM*, *t*(8) = −2.14, p>0.05; ANOVA: Phase X Group, *F*(1,8) = 16.57, p<0.005; Phase, *F*(1,8) = 50.33, p<0.001]. While the average response rate curve does suggest some degree of a shift occurred in the no-change group, a Bayesian analysis assessing evidence for a non-specific peak time shift in the no-change group revealed some evidence in favor of the null [Bayes Factor = 2.05]. Critically, the Bayesian analysis assessing evidence that the no-change group shift was equivalent to the change-group shift revealed decisive evidence in favor of the null [Bayes Factor = 1099.13]. As in the previous experiments, we did not observe CV changes in either group [Change-group, *M* = −14% ± 15% *SEM*, *t*(8) = −1.64, p>0.05; No-change group, *M* = 3% + /- 9% *SEM*, *t*(8) =.12, p>0.05].

Single-trial analyses revealed similar patterns, with a significant leftward shift in change-group start times and no reliable shift in the no-change group, although the difference between groups was only marginally significant in this case [Change group: *M* = −33% ± 5% *SEM*, *t*(8) = −3.27, p<0.05; No-change group: *M* = −4% ± 11% *SEM*, *t*(8) = −0.34, p>0.05; Change vs. No-change

group, $t(8) = -1.90$, p=0.094]. Stop times were reliably shifted in both groups, suggesting reward structure changes alone may have some impact on timed responding [Change group, $M = -46\% \pm 4\%$ SEM, $t(8) = -10.14$, p<0.001; No-change group, $M = -19\% \pm 5\%$ SEM, $t(8) = -4.12$, p<.005]. Importantly, the magnitude of the stop time shift was reliably larger in the change group than the no-change group [$t(8)=-2.44$, p<0.05].

Collectively, these results provide strong support for the proposal that cue-duration changes produce transfer, even when reward structure is controlled for. However, given the stop-time effect in the no-change group, we cannot rule-out that changes in overall reward structure can contribute to some degree of the shift in responding.

## Experiment 4a: violating the purported 'covariance assumption' blocks cross-cue transfer

According to the common cause hypothesis, observers rely on a 'covariance assumption' that the durations associated with different cues will change in the same direction. Therefore, if this expectation were explicitly violated (i.e. observers learn that the covariance assumption is not appropriate), the transfer should be attenuated or blocked. We tested this hypothesis next (see *Figure 5A and B*). First, we trained two groups of rats to associate 3 cues with a 4, 8, or 16 s duration, respectively. Then, in both groups, we increased the 16 s cue's duration to 32 s.

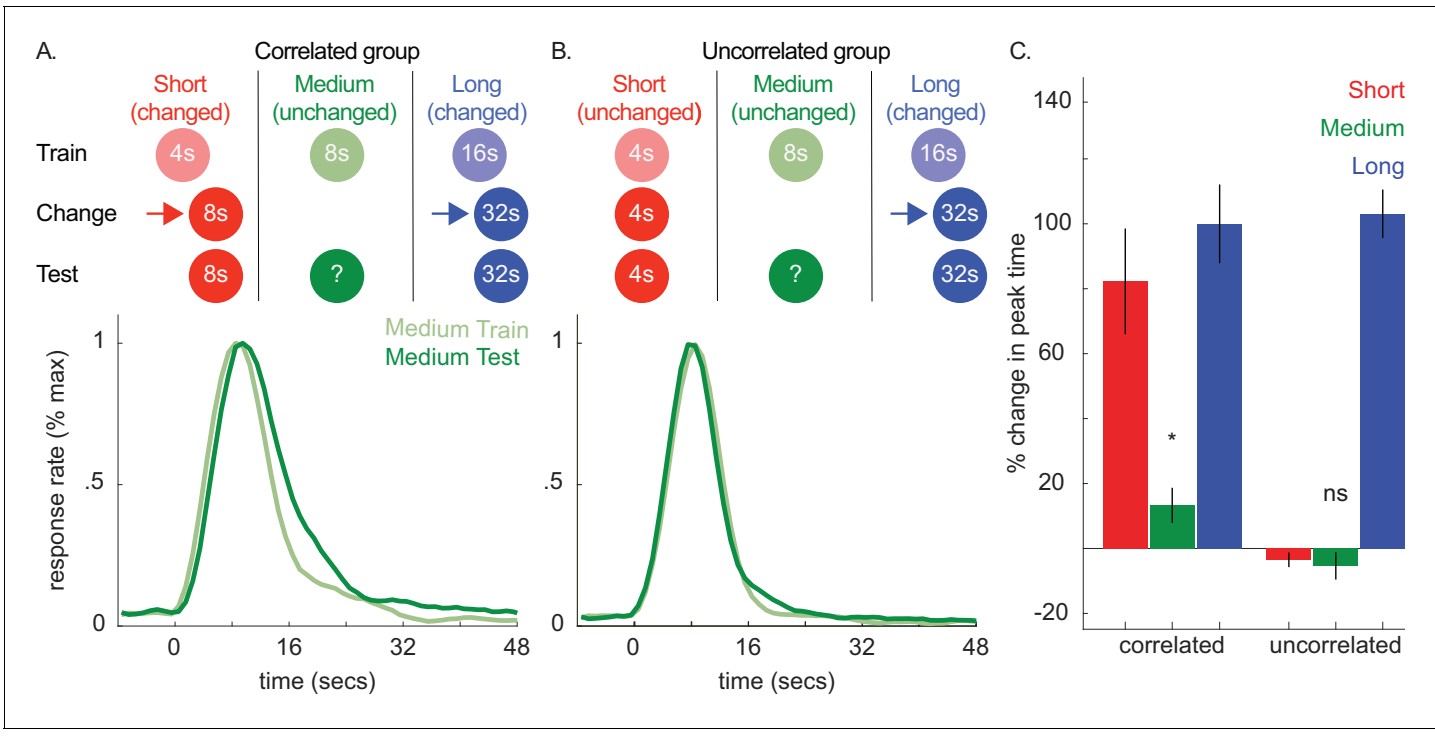

**Figure 5.** Top panels of A and B show designs used for the correlated (*n* = 5) and uncorrelated (*n* = 5) groups of Experiment 4a, respectively. Mean normalized response rate for the critical, medium cue during initial training and testing are plotted in the corresponding bottom panels. (C) shows percent change in peak times for all cues in each group during testing (±SEM).

DOI: https://doi.org/10.7554/eLife.38790.019

The following source data and figure supplements are available for figure 5:

**Source data 1.** Data used to generate *Figure 5*.
DOI: https://doi.org/10.7554/eLife.38790.022

**Figure supplement 1.** A and B show probe trial responding during initial training and testing for each cue in the correlated (*n* = 5) and uncorrelated (*n* = 5) groups of Experiment 4a, respectively.
DOI: https://doi.org/10.7554/eLife.38790.020

**Figure supplement 1—source data 1.** Data used to generate *Figure 5—figure supplement 1*.
DOI: https://doi.org/10.7554/eLife.38790.021

Importantly, in the 'correlated group', we also increased the 4 s cue's duration to 8 s. As both the 4- and 16 s cues' durations changed in the same direction, the covariance assumption should be upheld. Critically, in the other, 'uncorrelated' group, we held the 4 s cue's duration constant. This should violate the covariance assumption, as the 4- and 16 s cues did not change in the same direction. In both groups, we did not present the 8 s (i.e. unchanged) cue during this phase.

In a final test phase, we evaluated whether shifts to the 8 s cue would be absent or attenuated in the uncorrelated group. *Figure 5A and B* show response functions during probe trials both before and after the change phase in the correlated and uncorrelated groups, respectively. For clarity, only the peak functions for the critical, medium, cue are shown. However, *Figure 5C* shows the percent change in peak time between training and testing for all cues, and all curves are depicted in *Figure 5—figure supplement 1*.

Consistent with our prediction, transfer to the 8 s cue was observed in the correlated group and, more importantly, unreliable in the uncorrelated group [*Figure 5C*; Correlated group, $M = 13\% +/- 5\%$ *SEM*, $t(8) = 2.68$, $p<0.05$; Uncorrelated group, $M = -6\% \pm 4\%$ *SEM*, $t(8) = 1.01$, $p >. 05$; ANOVA: Phase X Group, $F(1,8) = 6.81$, $p<0.05$]. A Bayesian analysis assessing whether a peak time shift, per se, occurred in the uncorrelated group showed some evidence in favor of the null hypothesis [Bayes factor = 2.39]. Importantly, we found strong evidence that peak times did not shift in the uncorrelated group, relative to the alternative hypothesis that the peak-time shifts were equivalent to those seen in the correlated group [Bayes factor = 30.3]. In contrast to the previous experiments, overall CVs decreased slightly from training to testing [$M = -7\% \pm 2\%$ *SEM*, ANOVA: Phase, $F(1,8) = 10.96$, $p<0.05$]. Finally, single-trial analyses agreed with the peak time patterns, with start and stop time shifts being significant in the correlated group [Start, $M = 21\% +/- 7\%$ *SEM*, $t(8) = 3.44$, $p<0.01$; Stop, $M = 12\% +/- 4\%$ *SEM*, $ts(8) = 3.5$, $p<0.01$] and unreliable in the uncorrelated group [Start, $M = 1\% +/- 5\%$ *SEM*, $t(8) =.15$, $p>0.05$; Stop, $M = -3\% \pm 4\%$ *SEM*, $t(8) = -0.78$, $p>0.05$].

Collectively, these results indicate that violating the purported covariance assumption blocks directional transfer to the unchanged cue.

## Experiment 4b: assessing within-mode generalization on transfer

While these data support the common cause hypothesis, there was one peculiarity in the results. Specifically, the magnitude of the shift seen in the correlated group of Experiment 4a ($M = 13\%$) appears to be weaker than that seen in the 16-to-32 group of Experiment 1 ($M = 59\%$). Given that the covariance assumption was explicitly confirmed in the correlated group, but not in the 16-to-32 group of Experiment 1, one would expect that the transfer would be magnified in the correlated group or, at the very least, on-par with the 16-to-32 group.

We hypothesized that a seemingly mundane aspect of the design used in Experiment 4a might account for this difference. In that experiment, the short and long cues were both in the auditory modality, whereas the medium cue was a light. Therefore, we wondered whether having the two shifted cues (short and long) being presented through the same mode (auditory) might have encouraged rats to categorize the cues by modality, thereby weakening any transfer to the medium (visual) cue.

To assess this possibility, we ran a follow-up, which was essentially a counter-balanced version of the prior experiment. We used the same experimental design as Experiment 4a (see *Figure 6*), but the short and long cues were visual and auditory stimuli, respectively, and the medium cue was auditory. We reasoned that, for the correlated group, having the two shifted cues be presented through distinct modes (visual and auditory) would eliminate any within-mode categorization and, thereby, boost the transfer to the medium cue.

Replicating our prior effect, medium cue peak time shifts were reliable in the correlated group and unreliable in the uncorrelated group [*Figure 6C*; *Figure 6—figure supplement 1*; Correlated group 8 s cue shift, $M = 53\% +/- 22\%$ *SEM*, $t(8) = 3.5$, $p<0.05$; Uncorrelated group 8 s cue shift, $M = 5\% +/- 3\%$ *SEM*, $t(8) =.56$, $p >. 05$; ANOVA: Phase X Group, $F(1,8) = 6.16$, $p<0.05$]. The Bayesian analysis assessing whether a non-specific shift occurred in the uncorrelated group showed strong evidence in favor of the null [Bayes factor = 11.18]. Furthermore, we found strong evidence that peak times did not shift in the uncorrelated group, relative to the alternative that the peak-time shifts were equivalent to the correlated group [Bayes factor = 40.97]. We found no CV changes during testing [Correlated, $M = 2\% +/- 7\%$ *SEM*, $t(8) =.12$; $p>0.05$; Uncorrelated, $M = 16\% +/- 8\%$ *SEM*, $t(8) = 2.07$, $p>0.05$]. Finally, single-trial analyses agreed with the peak time patterns, with

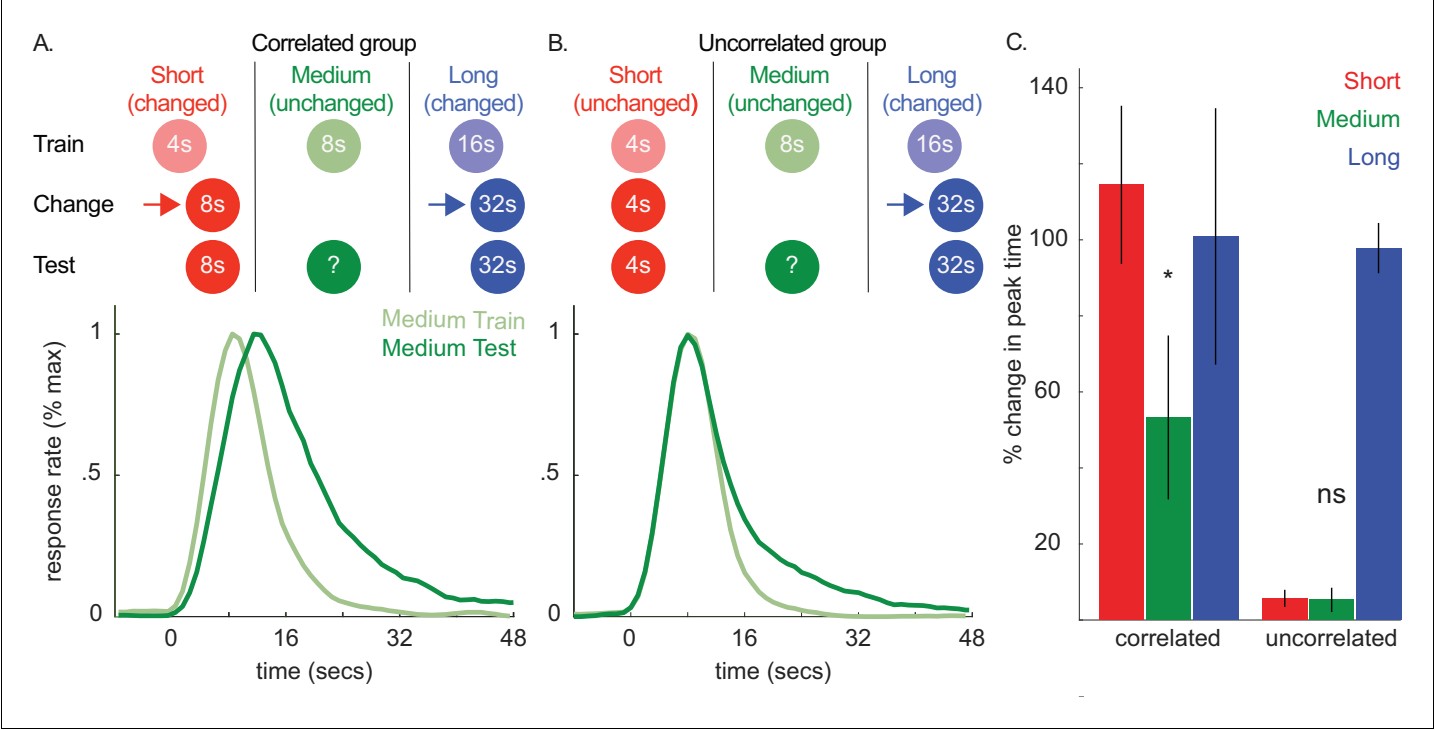

**Figure 6.** Top panels of (A) and (B) show designs used for the correlated (*n* = 5) and uncorrelated (*n* = 5) groups of Experiment 4b, respectively. Mean normalized response rate for the critical, medium cue during initial training and testing are plotted in the corresponding bottom panels. (**C**) shows percent change in peak times for all cues in each group during testing (±SEM).

DOI: https://doi.org/10.7554/eLife.38790.023

The following source data and figure supplements are available for figure 6:

**Source data 1.** Data used to generate *Figure 6*.

DOI: https://doi.org/10.7554/eLife.38790.026

**Figure supplement 1.** (A) and (B) show probe trial responding during initial training and testing for each cue in the correlated (*n* = 5) and uncorrelated (*n* = 5) groups of Experiment 4b, respectively.

DOI: https://doi.org/10.7554/eLife.38790.024

**Figure supplement 1—source data 1.** Data used to generate *Figure 6—figure supplement 1*.

DOI: https://doi.org/10.7554/eLife.38790.025

rightward shifts in start and stop times being significant in the correlated group [Start, *M* = 63% + /- 30% *SEM*; Stop, *M* = 47% + /- 19% *SEM*; *t*s(8) > 3.5, *p*s <0.01] and unreliable in the uncorrelated group [Start, *M* = 6% + /- 4% *SEM*; Stop, *M* = 15% + /- 7% *SEM*; *t*s(8) <.4, *p*s >0.05].

Importantly, the magnitude of the peak-time shift in the correlated group was roughly four times larger in this experiment (*M* = 53%), compared to the prior experiment (*M* = 13%), and more in line with the shift observed in the 16-to-32 group of Experiment 1 (*M* = 59%).

## Experiment 5: cross-cue transfer is context-dependent

We hold that the mechanism mediating the transfer evolved to allow individuals to benefit from the frequently co-varying causal contributions of the environment. We tested a final prediction based on the purported functional purpose of this mechanism.

To illustrate, we will again use the metaphor of an animal foraging in an area that contains two plant types. When foraging through plant A, it finds food every 8 s, whereas searching through plant B yields food every 16 s. Under stable conditions, the intervals associated with either plant type will remain constant. However, during a drought, both plants would become depleted, causing their inter-food intervals to increase. According to the common cause hypothesis, if the animal learns that plant B's inter-food interval increased, it will expect that plant A's interval also increased. Our results support this notion.

However, what would happen if the animal left the drought impacted area and travelled to a new location that also contained plant types A and B. Would the assumption that plant A's interval has changed be appropriate in this new context? Furthermore, should the animal expect that plant B's interval changed, as it did in the drought impacted area? As an extreme example, if the two locations were located on different continents, a drought affecting one should have little relevance to the other.

The essential point posed in this example is that, in the environment, common causal factors that impact the durations associated with different temporal cues (e.g. drought) should be context-specific. We tested this prediction in our next experiment (see *Figure 7A and B* for design). First, we trained rats to associate a tone and light with either an 8- or 16 s duration in one context. Then, we moved them into a novel context, and trained them that the 16 s cue now predicted reward after 32 s. Finally, we evaluated how they would respond to both cues when tested in the original training context versus the novel context (i.e. no feedback was given for either cue). If the common cause hypothesis is correct, any transfer should be eliminated or attenuated when rats are tested in the original training context, relative to the novel context.

This hypothesis was confirmed, as shown in *Figure 7*, which compares responding during initial training to testing in the original training (i.e. no-change) context or the novel (i.e. change) context

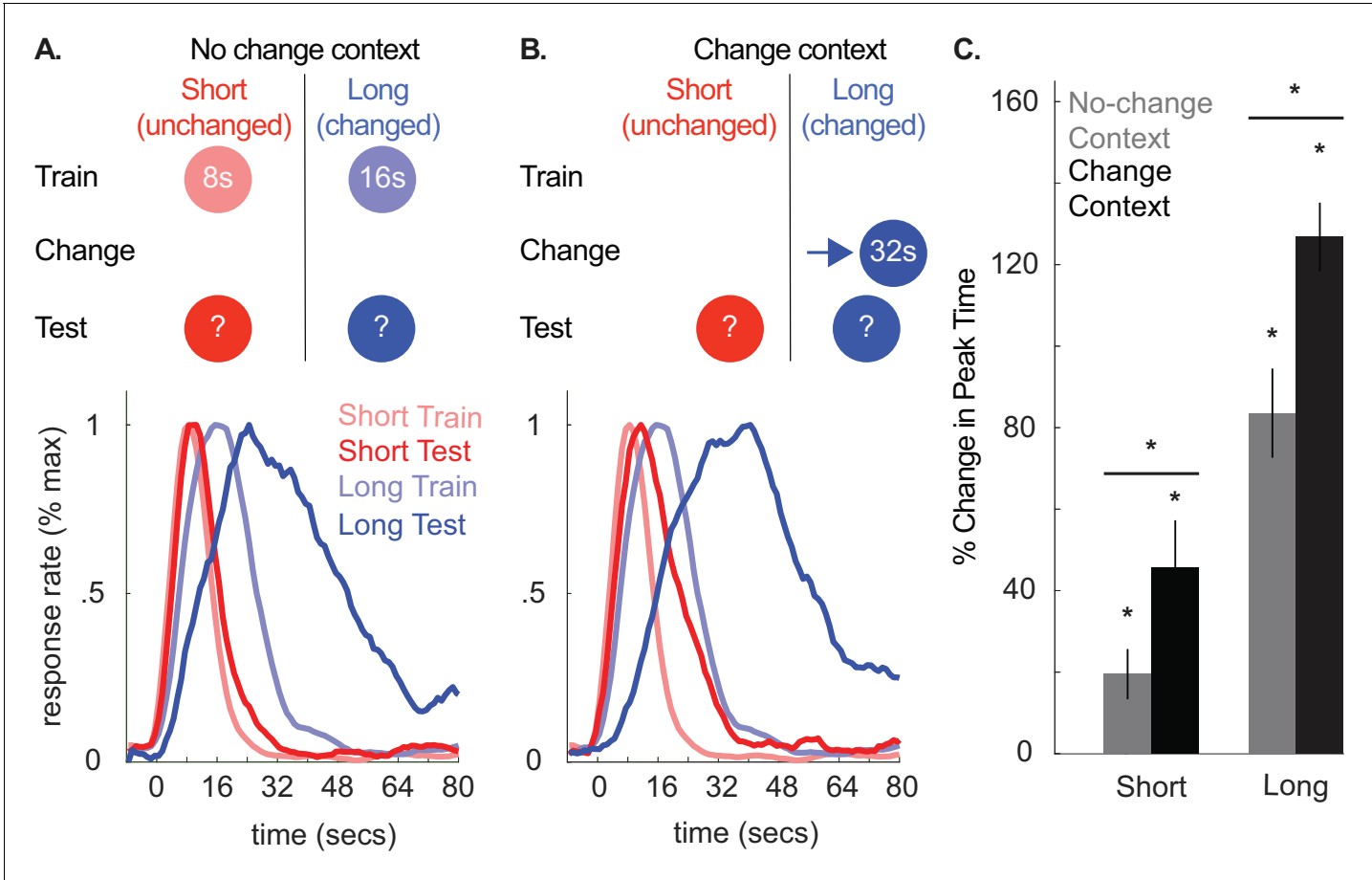

**Figure 7.** Top panels of (A) and (B) illustrate the design used in either context during Experiment 5 (*n* = 20). Panels below each schematic show corresponding normalized response rates during training and during testing in either context. (C) shows percent change in peak time in either context during testing, relative to initial training (±SEM). Stars indicate significance.
DOI: https://doi.org/10.7554/eLife.38790.027

The following source data is available for figure 7:

**Source data 1.** Data used to generate *Figure 7*.
DOI: https://doi.org/10.7554/eLife.38790.028

(*Figure 7A and B*, bottom panels, respectively). *Figure 7C* shows percent change in peak times in both contexts during testing, relative to initial training. Short, unchanged cue responding shifted in both contexts. Critically, this transfer was attenuated during testing in the no-change context [No-change context shift, $M$ = 19% + /- 6% *SEM*, $t(19)$ = 3.01, p<0.01; Change-context shift, $M$ = 46% + /- 12% *SEM*, $t(19)$ = 4.20, p<.005; No-change context shift vs. Change context shift, $t(19)$ = −2.70, p<0.05; ANOVA: Phase X Test Context, $F(2,32)$ = 11.16, p<0.001; Phase, $F(2,32)$ = 65.25, p<0.001; Cue, $F(1,16)$ = 392.28, p<0.001; Phase X Cue, $F(2,32)$ = 37.33, p<0.001]. Moreover, responding to the long, changed cue also shifted leftward in the no-change context relative to the peak-times in the change context [$M$ = 84% + /- 11% *SEM* vs. 127% ± 8% *SEM* from training, respectively; $t(19)$ = −3.31, p<.005], supporting the notion that the temporal expectations for both cues change in a correlated manner.

Minor CV effects were also observed. Specifically, all CVs were slightly lower during the second day of testing (i.e. when tested in the context they were not tested in during the first day of testing) [$M$ = −12% ± 4% *SEM*, ANOVA: Phase, $F(2,32)$ = 4.00, p<0.05] and also lower overall for the long cue [$M$ = −7% ± 3% *SEM*, ANOVA: Cue, $F(1,16)$ = 5.00, p<0.05]. No context effects were present [$F$s <1.3, $p$s >0.05].

Single-trial analyses were broadly consistent with the peak time results. Specifically, short-cue start times were rightward shifted in both contexts, relative to training [No-change context, $M$ = 38% + /- 10% *SEM*, $t(19)$ = 3.60, p<.005; Change-context, $M$ = 68% + /- 21% *SEM*, $t(17)$ = 3.35, p<.005]. Short-cue stop times were also rightward shifted in the change context, relative to training [$M$ = 46% + /- 12% *SEM*, $t(17)$ = 3.80, p<.005], and notably, not reliably shifted in the no-change-context [$M$ = 10% + /- 7% *SEM*, $t(19)$ = 1.10, p>0.05]. While start times during testing did not differ between the two contexts [$t(17)$=1.60, p>0.05], this comparison was significant for stop times [$t(19)$ =3.99, p<.005].

Start and stop times for the long cue followed a similar pattern. Specifically, start times were rightward shifted in both contexts, relative to training [No-change context, $M$ = 189% + /- 26% *SEM*, $t(19)$ = 7.49, p<0.001; Change-context, $M$ = 226% + /- 25% *SEM*, $t(18)$ = 11.84, p<0.001], although the difference between change-context and no-change context at test was unreliable [$t(18)$=−1.00, p>0.05]. Finally, stop times were rightward shifted in both contexts, relative to training [No-change context, $M$ = 50% + /- 8% *SEM*, $t(19)$ = 5.86, p<0.001; Change context, $M$ = 89% + /- 8% *SEM*, $t(18)$ = 13.49, p<0.001], and this difference was greater in the change context than the no-change context [$t(18)$=−3.9, p<.005].

## Discussion

### Summary

We proposed and tested predictions made by the common cause hypothesis. This account emphasizes that, in a real-world setting, the durations associated with distinct temporal cues will often covary due to a common underlying cause. We evaluated whether a mechanism has evolved that allows observers to utilize this statistical property in order to adaptively recalibrate expectations. Consistent with this, we found that, when the duration associated with one temporal cue changes, the temporal expectations associated with other cues shift in the same direction (Experiments 1–3). Furthermore, in Experiment 4, we found that, when the durations associated with multiple temporal cues do not covary, the transfer is absent. Finally, in Experiment 5, we confirmed that this transfer is context-dependent. These results have important implications within and beyond the field of timing.

### Model-free vs. Model-based mechanisms?

The common cause hypothesis focuses on the function that the transfer we document here serves to the organism in question. However, it does not propose a theoretical mechanism by which the transfer occurs. A variety of mechanisms can potentially account for our results. The topic can broadly be framed in terms of the contrast between 'model-free' and 'model-based' behavior (*Daw et al., 2005*; *Doll et al., 2012*; *Gläscher et al., 2010*; *Petter et al., 2018*).

In model-based behavior, actions are guided by cognitive models of the sequence of events and choices that lead to desired outcomes (e.g. reward). With respect to the transfer we observe, rats in our studies could have relied on a statistical model of the expected covariance between the different

entrained intervals and used this to guide responding to the unchanged cue. They could even have explicitly inferred that the changed-cue's duration shifted due to a common cause that operated on all cues, as contemporary data suggest that rodents engage in causal reasoning, make inferences about latent variables in the environment, and rely on covariance computations during conditioning experiments (*Blaisdell et al., 2006*; *Cheng and Novick, 1992*; *Gershman et al., 2015*; *Starkweather et al., 2017*; *Wilson et al., 2014*).

However, dissociating whether rats truly engage in causal and/or statistical inference or simply act *as if* they are doing so is a difficult problem. For example, 'model-free' processes could have guided the transfer. In this behavioral mode, simple stimulus-response associations guide behavior, rather than more complex cognitive models. With respect to our data, a particularly important consideration is how simple extinction and reinforcement mechanisms could produce the transfer.

Specifically, in our tasks, changing a cue's duration entailed extinguishing responses emitted at the old reinforced time and reinforcing responses emitted at the new time. If these response-level extinction and reinforcement processes carried over to unchanged cue trials, responding would shift in the direction of the changed cue's new duration. To assess this, we recommend using a design in which each cue is associated with a different action-outcome contingency (e.g., Cue 1: 8 s/nosepoke; Cue 2: 16 s/lever press). During the change phase, model-free processes would operate on different actions. Therefore, under this account, one would not expect the transfer to occur.

## Implications for theories of timing

As far as we are aware, no traditional timing theory can readily (i.e. without modification) account for our findings. For example, established models of timing at both the cognitive (*Gibbon et al., 1984*; *Killeen and Fetterman, 1988*; *Kirkpatrick, 2002*; *Machado et al., 2009*) and neural level (*Matell and Meck, 2004*; *Simen et al., 2011*) typically assume that the temporal expectations associated with different cues are independent of one another. That is, a change in one cue-duration relationship should have no impact on performance for the others an observer has learned. Our results clearly challenge this assumption.

This is not to say that timing models could not be modified to account for our findings. Our primary point is that the information that guides responding to a temporal signal may be far more complex than often assumed, being guided by a host of variables that go beyond a given cue-duration relationship. This has been proposed previously and is supported by work showing that rodents take non-temporal factors into account when guiding responses in time, such as the probability of receiving reward after a given duration (*Balci et al., 2009*; *De Corte and Matell, 2016a*; *Gür and Balcı, 2017*). Furthermore, during Pavlovian conditioning, the learning-rate for a cue is determined by the relative relationship between the duration separating cue-onset and reward-delivery and the interval separating successive rewards, rather than the cue-reward interval alone (*Balsam and Gallistel, 2009*; *Gallistel and Balsam, 2014*; *Gallistel and Gibbon, 2000*).

## ACT-R pooled memory model

There is one model that is important to discuss, as it was able to predict aspects of the data where traditional models failed. Specifically, *Taatgen and van Rijn (2011)* recently proposed a modified clock-memory model based on the well-established ACT-R cognitive architecture, a theory that has enjoyed wide success at explaining cognitive phenomena across a variety of domains (*Anderson, 1996*). In their model, the 'temporal memory' associated with a cue is determined by a pool of recently experienced intervals. Importantly, this pool consists of intervals associated with all cues an observer has experienced in the recent past. When a cue is presented, the observer retrieves a blend of the durations contained in this pool. More weight is given to the intervals associated with the cue in question, causing the final estimate to fall close to its true duration. However, the intervals associated with other cues also bias the retrieved estimate, due to memory interference. In support of this, they documented cross-cue transfer in humans, similar to the transfer we observed.

Our results can inform this model in several respects. First, Experiment 4 suggests that transfer is not inevitable when the duration associated with one cue changes. Rather, duration-change covariance determines whether reliable transfer occurs, which is currently not incorporated in the model. Second, Experiment 5 suggests that the physical context in which a memory is retrieved strongly modulates cross-cue transfer. Interestingly, in their model, temporal memories are treated as

declarative knowledge, and this proposition was recently supported by work in mice showing that temporal and episodic memory may be highly inter-related processes (*Kheifets et al., 2017*). Work in other domains has shown that declarative memory retrieval is heavily influenced by physical context (*Bilodeau and Schlosberg, 1951*; *Godden and Baddeley, 1975*; *Smith and Vela, 2001*), and ACT-R has been equipped to deal with such contextual variables in the past (*Anderson, 1996*; *Anderson et al., 2004*). Therefore, adding a context-component may be straightforward.

Finally, there is a conceptual problem with applying this theory to our data. Specifically, according to their model, the transfer occurs due to memory interference. In some of our experiments, unchanged cue responding shifted by as much as 60% (e.g., the 16-to-32 group of Experiment 1). Pragmatically, if memory interference were responsible for such a dramatic shift, one would have to assume that the interval timing system is incredibly delicate and potentially question what utility it has in a real-world context. The common cause hypothesis does not suffer from this critique, as the transfer serves a function.

Exploring these modifications could potentially provide a bridge between the timing field and research in other cognitive domains in which ACT-R is more often utilized. Importantly, the theory could provide an avenue for mathematically formalizing the common cause hypothesis, which, at present, can only predict whether or in what direction transfer will occur.

## Future directions and beyond timing

This work opens several avenues for future research on the effects documented here. For example, an important question is whether the predictions of the common cause hypothesis will extend beyond the domain of time alone. We feel that this is highly likely within the context of both prior work and the assumptions of the common cause hypothesis.

For example, evidence suggests that dimensions such as time and number rely on common-coding mechanisms (*Brannon et al., 2007*; *Gallistel and Gelman, 2000*; *Leibovich et al., 2017*; *Meck and Church, 1983*; *Riemer et al., 2016*). Furthermore, statistical and/or causal relationships among different environmental cues should not occur within the dimension of time alone. In fact, this can be clearly illustrated from the foraging example that initially inspired the common cause hypothesis. Specifically, during times of increased rainfall, plants might produce more and/or larger fruit than normal, leading to covariance among size and number, in addition to duration. Therefore, one would expect transfer within these dimensions as well.

A potentially more interesting point to note is that, in this example, duration, size, and number will also co-vary *across* dimensions (e.g. a higher amount of food will shorten a plant's inter-food interval). Therefore, from the common cause hypothesis, one might expect transfer across these dimensions as well. In support of this, many have documented correlated interactions between the dimensions of time, space, and number (*Merritt et al., 2010*; *Rugani and de Hevia, 2017*; *Walsh, 2003*), such as when larger stimuli are perceived as lasting a longer amount of time than smaller stimuli (*De Corte et al., 2017*). Investigating this proposal might reveal that our results relate to a more general computational principle used for generating expectations of any kind, rather than being a component of the timing system alone.

Another direction would be to investigate how the cue interactions observed here relate to those seen in other areas of learning and memory research. For example, the 'peak-shift' effect, wherein the generalization gradient for a reinforced stimulus is systematically shifted away from a non-reinforced stimulus, is a prominent area of conditioning research (*Akins et al., 1981*; *Cheng et al., 1997*; *Lynn, 2010*; *Siegel, 1967*). Typically, the degree of generalization is determined by their perceptual similarity (e.g. relative brightness). In our study, we chose highly distinct cues whenever possible (often in different modalities) to minimize interpretations related to perceptual generalization. However, the sensory similarity of the two cues may modulate the transfer (as suggested by Experiment 4). Furthermore, our findings may relate to the 'acquired equivalence' effect, whereby stimuli that predict the same outcome are treated as more similar than stimuli that predict different outcomes (*Coutureau et al., 2002*; *Honey and Hall, 1989*). Further investigations could evaluate how well these phenomena relate to the transfer we observe.

Finally, an important question is whether the transfer follows an optimal policy. As noted above, the common cause hypothesis can primarily predict whether transfer will occur and in what direction. However, we also propose that the transfer serves an adaptive function. If this were the case, one would expect that the degree of transfer would follow a systematic optimization principle, based on

the expected covariance between the cues. To address this, we recommend explicitly training the animals on what degree of temporal covariance to expect between cues by including phases in which all cues are reinforced at new times and the degree of the shifts follows a programmed covariance relationship. If only a single cue were shifted during a final change phase, one could evaluate whether responses to the remaining cue(s) follow the experimentally chosen covariance policy.

## Materials and methods

We begin with a concise overview of our procedures and analyses. We then give more in-depth methods for each experiment and analysis in the 'detailed experimental procedures' section below. Datasets and all functions used for analysis are available as source files associated with the manuscript, both in a compiled (i.e. data/code for all experiments) and figure-specific manner.

### Ethics statement

All procedures accorded with Villanova University's Animal Care and Use Committee guidelines (IACUC, protocol 1880) and the Declaration of Helsinki.

### General method

All rats were initially trained on the peak interval procedure under modest food deprivation (85 – 90% of free-feed weight) and in standard operant conditioning chambers, described previously (*De Corte and Matell, 2016b*). Training progressed through three phases. During an initial 'training phase', trials began with the presentation of a cue that predicted reward availability after a certain duration elapsed. Different cues were used, each associated with a different delay to reward (e.g. tone-8s/light-16s). Rats were free to respond at any point when the cue was presented. However, reward was only delivered for a response after the cue's duration had passed. When a rewarded response occurred, the cue terminated and reward was delivered. Probe trials (30% of trials within a session for each cue) were also included in which the cue was presented for much longer than normal (3–4 times the length of the longest duration in effect), and no reward was provided. Once trained, a 'change phase' was implemented. During this phase, the duration associated with one of the cues was changed (i.e. changed cue), and one of the cues was no longer presented (i.e. unchanged cue). Exceptions include the control group of Experiment 1 in which we presented one cue without altering its duration (i.e. 'exposed' cue), yet still omitted the other cue (i.e. 'unexposed' cue). In Experiment 3's change phase, the short cue's duration in the 'change group' was shifted to 4 s, whereas in the no-change group, the cue's duration remained at 8 s. Additionally, a novel cue was added during this phase, signaling reward availability at 8 s (change group) or 4 s (no-change group). In Experiment 4a and 4b, we either changed two cues' durations (correlated group) or one cue's duration and left another unchanged (uncorrelated group). Finally, in all experiments, a 'test phase' was instituted. This was identical to the change phase. However, we reintroduced the cue that was omitted in the previous phase (20% of trials within a session). No feedback was given during unchanged cue trials. Furthermore, in Experiment 5, all trials were probes, as evaluating shifts in responding for both cues were relevant in that case.

### Data analysis

We grouped each rat's responses into 1 s bins. Then, for each cue, we computed average response rate as a function of time within a trial. These 'peak functions' were fit (Matlab curve fitting package, Cambridge, MA) using a five-parameter Gaussian function: $Y = B + S * exp((-1/2) * (abs(T - PT)/SP)^K)$ (*Matell et al., 2016*; *Swanton et al., 2009*). B represents the baseline, S is a scale parameter, T is time, PT is the mean, SP is the spread, and K allows the function to fit peaks with varying degrees of kurtosis. The mode of the distribution was taken as the peak time. We also computed start and stop times during single trials as described previously (*Church et al., 1994*; *Gibbon and Church, 1992*). Briefly, we fit three flat lines to the rate of responding during individual trials (first low, second high, third low). The left and right sides of the higher line were taken as the start and stop time, respectively.

We compared data from initial training and the test phase with mixed-model ANOVAs. Cue, Phase (training vs. test), and Context (i.e. for Experiment 5; change-context vs. no-change context) served as within-subjects factors. Group and, where applicable, Modality-Duration subgroup (e.g.

tone-short/light long vs. tone-long/light short) were used as between-subjects factors. The Modality-Duration relationship did not exert reliable effects in cases where these factors were counterbalanced within an experiment (the four groups in Experiments 1–2). To simplify statistical reporting, these results are not detailed. Planned comparisons were conducted using paired and independent samples t-tests. Critical interactions were evaluated with simple effects. To clearly demarcate planned comparisons from ANOVA statistics, these results are reported using t-values (i.e. taking the square root of the F-statistic and listing the corresponding degrees of freedom). In some cases, we were interested in whether transfer occurred in one group, but not in another (Experiments 1, 3, and 4). To assess evidence in favor of the null hypothesis for the latter case, we used two Bayesian analyses. One utilized an 'uninformative' or 'incremental' alternative prior (*Gallistel, 2009*) and provided evidence regarding whether a shift in peak times occurred, per se. The null prior was the marginal likelihood distribution of peak times from the group in question, linearly shifted to have a mean of zero. The second analysis was more specific and compared data from the group that was expected to show a shift to the one that was not. The null prior was the same as described above. However, the alternative prior was the marginal likelihood function of peak times from the group that was expected to show a shift.

## Detailed experimental procedures

### Subjects and apparatus

100 naïve, male, Sprague-Dawley rats, who were 2 – 3 months of age at the start of each experiment were used as subjects. Rats were housed in pairs, given free access to water, but food restricted to 85 – 90% of their free-feed weights, adjusted for growth. Colony room lights were set to a 12 hr light-dark cycle. Rats were randomly assigned to experimental groups, and sample sizes for each experiment (10 at a minimum, with a minimum of 5 rats per subgroup) were determined based on prior studies in which a reliable peak time shift of at least 10–20% was assessed (*De Corte and Matell, 2016a*; *Gooch et al., 2007*; *Matell and Meck, 2004*; *Matell et al., 2006*). Rats in Experiments 1, 2, and 5 were run in 2 hr sessions during the light cycle, while rats in Experiment 3 – 4 were run for 12 hr sessions during their dark cycle to accelerate completion of the experiment, due to the increased complexity of training (see below). Subjects were trained and tested in standard operant conditioning chambers (30.5 × 25.4×30.5 cm; Coulbourn Instruments, Allentown, PA). The chambers' left and right walls consisted of ventilated Plexiglas. The front, back, and top sides of the chambers were composed of aluminum. Floors consisted of an array of stainless steel bars. Three nosepoke apertures, equipped with photobeam detection circuits, were located on the back wall. An 11 lux houselight and seven-tone audio generator, set to produce 90 dB tones, were located along the back wall of the chambers as well. A pellet dispenser on the front wall was used to deliver 45 mg grain pellets into a food magazine (Bio-Serv, Flemington, NJ). A fan (60 dB) provided ventilation. Stimulus control and data acquisition were accomplished using a standard operant-conditioning control program (Graphic State 4, Coulbourn Instruments, Allentown, PA), with a temporal resolution of 1 ms. Prior to behavioral training, rats were given approximately twenty 45 mg grain pellets in their home cage to acclimate them to the reward used in the chambers.

### Procedure

All experiments progressed through five stages: nosepoke training, fixed-interval training, peak-interval training, a 'change-phase', and a 'test-phase'. Each experiment was performed only once.

### Nosepoke training

During nosepoke training, rats were trained to insert their snouts into the center nosepoke in order to earn reward. Each response (snout insertion) resulted in reward delivery. To prevent multiple, rapid responses from potentially jamming the pellet dispenser, reward delivery was followed by a 1 s timeout, during which further responses did not activate the feeder. Training continued until rats received 60 rewards for two consecutive sessions.

### Fixed-interval training

During fixed-interval training, trials began with the onset of an individual cue (randomly determined on each trial; specific cues used detailed in individual experiment description). Each cue was

associated with a distinct delay to reward availability (e.g. tone-8s/light-16). During all trials, the cue remained on until the first nosepoke response following the passage of the corresponding fixed interval, at which point reinforcement was delivered. There was no programmed consequence for responding before the fixed interval had elapsed. Once a reinforced response occurred, the cue terminated and a uniform, 60 – 80 s inter-trial interval commenced.

### Initial peak-interval training phase
Peak-interval training was identical to FI training with one exception. In addition to fixed-interval trials in which reward was delivered, probe trials (30% of trials for each cue) were introduced, in which reward was omitted and the stimulus was presented for 3 – 4 times the longest duration in effect (e.g. 48 – 64 s if the longest duration was 16 s). Unlike rewarded trials, probe trials terminated independently of responding.

### Change-phase
The change phase was similar to peak-interval training. However, one of the cues was omitted, and the durations associated with the other cue(s) were either altered or left unchanged. The specific manipulations are specified in the individual experiments' procedural descriptions below.

### Test-phase
The test-phase was identical to the change-phase except we introduced probe trials (20% of all trials within a session) in which the cue that was omitted during the change phase was reintroduced. As 'unchanged-cue' trials were always probes, rats were never given feedback about when to respond. The critical question was whether the time of responding for the 'unchanged cue' shifted between the peak-interval (i.e., training) phase and the test phase and, if so, in what direction.

### Individual experiment descriptions
#### Experiment 1
Rats ($n = 30$) were trained that a 1 kHz tone and the houselight were associated with 8 s and 16 s durations until availability of reinforcement. During the change phase, the interval associated with one of the cues was changed, and the other cue was not presented. For half the shift rats ($n = 10$), the interval associated with the short, 8 s cue was changed to 4 s. For the other half of the shift rats ($n = 10$), the interval associated with the long, 16 s cue was changed to 32 s. In the 16-to-32 group, probe trials were extended to 96–128 s to provide sufficient time to assess peak responding. Control rats ($n = 10$) were trained and tested identically to those in the 8-to-4 group above. However, during what would have been the change phase, the short cue's duration was not altered (i.e. remained at 8 s). The modality-duration relationship was counterbalanced in all groups. As discussed further below, response rates during extinction trials were low and more variable than normal during testing. This was particularly problematic in the 8-to-4 group, with some rats showing a brief burst of responding late into a probe trial, presumably in anticipation of trial end. To prevent these trials from interfering with our measures of peak responding, we excluded trials for this group in which the first response did not occur before 32 s had elapsed (i.e. twice the length of the longest duration in effect). This restriction is highly conservative, relative to other methods used to filter non-timed responses from probe trial data (e.g. *Church et al., 1994*; *De Corte and Matell, 2016a*).

#### Experiment 2
Rats ($n = 10$) were trained identically to Experiment 1. However, during the change phase, the short cue's interval was lengthened to 12 s. The modality-duration relationship was counterbalanced.

#### Experiment 3
Rats ($n = 10$) were initially trained to associate a continuous, 1 kHz tone with an 8 s (short) fixed interval schedule and a beeping (10 Hz - 50 ms on, 50 ms off) 4 kHz tone signaled a 16 s (long) fixed-interval schedule. During the change phase, rats were split into two groups ($n = 5$ each). In the first 'change' group, the 8 s cue's duration was decreased to 4 s. We also introduced a novel houselight cue that was associated with an 8 s fixed-interval schedule. In the second 'no-change' group, the 8 s

cue's duration was held constant, whereas the novel houselight cue was associated with a 4 s duration.

## Experiment 4

In Experiment 4a, rats (*n* = 10) were trained with three cue-interval relationships. A continuous 1 kHz tone signaled a 4 s (short) fixed interval schedule, the houselight signaled an 8 s (medium) fixed-interval schedule, and a beeping (10 Hz - 50 ms on, 50 ms off) 4 kHz tone signaled a 16 s (long) fixed-interval schedule. As in previous experiments, 70% of all trials for each cue were reinforced, and the remaining 30% were non-reinforced. During the change phase, all rats were retrained that the long (beeping tone) cue was lengthened from 16 s to 32 s. Half of the rats were also retrained that the short cue (continuous tone) was lengthened from 4 s to 8 s (correlated group; *n* = 5), whereas in the other half of the rats, the short cue's duration was maintained at 4 s (uncorrelated group; *n* = 5). During the test phase, we presented the two auditory cues as described above, and presented the medium cue (i.e. houselight) as probe trials only (20% of trials during a session). Experiment 4b was identical. However, the short and long cues were the continuous tone and light, respectively, and the medium cue was the beeping tone. There was a further minor difference due to a procedural error. Specifically, probe trials during the training phase were initially 48 – 64 s and then extended to 96 – 128 s during the change phase, whereas in Experiment 4a, probe trials were 96 – 128 s throughout the experiment.

## Experiment 5

Rats (*n* = 20) were trained identically to Experiment 1, except that rats were trained in two different contexts. In addition, the probe lengths for all trials were 96 – 128 s. While the composition of the operant chambers were identical for all rats, half the rats (*n* = 10) were trained in a lab room on the 4th floor, with clay based cat litter (Arm and Hammer Double Duty) serving as 'bedding' underneath the metal bars of the operant chamber floor (Context A). These rats were moved via a wheeled cart down a hallway to and from a 4th floor colony room to the testing room on each session. The other half of the rats (*n* = 10) were housed in a colony room on the 3rd floor, and testing occurred in an adjacent room with wood shaving underneath the chamber floor (Context B). During the change phase, the location of all rats was switched (i.e. rats trained in Context A were retrained in Context B and vice-versa). Furthermore, all rats were presented with only the long cue, and the interval until reinforcement availability was lengthened to 32 s. The test phase was two sessions long. One session took place in the rats' original training context, whereas the other session took place in the novel context in which rats had learned the long cue's new duration. During both sessions, rats were presented with both the short and long cues as probe trials only, and each cue was presented with an equal probability. The training/novel context testing order was counter-balanced.

### Parametric analyses

Each rat's responses were grouped into 1 s bins and mean response rate as a function of time within a trial was computed. These 'peak functions' were fit (Matlab curve fitting package, Cambridge, MA) using a five-parameter Gaussian function: $Y = B + S * \exp((-1/2) * (abs(T - PT)/SP)^K)$, as used previously (*De Corte and Matell, 2016b*; *Swanton et al., 2009*). B represents the baseline rate of responding, S is a scale parameter, T is time, PT is the mean, SP is the spread, and K allows the function to fit peaks with varying degrees of kurtosis. The mode of the distribution was taken as the peak time.

We also computed start and stop times during single trials as described previously (*Church et al., 1994*). Briefly, we fit three flat lines to responding during individual trials (first low, second high, third low). The left and right sides of the higher line were taken as the start and stop time, respectively. Given that the critical data of interest (i.e., unchanged cue responding) came from extinction trials, single-trial behavior was variable, as noted previously (*Roberts and Gharib, 2006*), and response rates decreased rapidly due to a lack of reinforcement. To prevent non-temporally controlled responses from being spuriously registered as start and stop times, we only analyzed trials in which at least three responses were emitted and the obtained start and stop times occurred before and after the obtained peak time, respectively (*Church et al., 1994*; *Gibbon and Church, 1992*).

We compared peak, start, and stop times from initial peak-interval training and the test phase with a mixed-model Analysis of Variance (ANOVA) implemented in SPSS. We matched the number of training and test sessions for data analysis. Cue (short vs. long), Phase (initial peak-interval training vs. test-phase), and Context (Experiment 5; no-change vs. change) served as within-subjects factors. Where applicable, Group (e.g., correlated vs. uncorrelated groups of Experiment 4a/b) and the modality-duration subgroup (e.g., 8-to-4, 16-to-32, 8-to-8 groups of Experiment 1 and the 8-to-12 group of Experiment 2) served as between-subjects factors. We did not observe effects related to Modality-Duration subgroups, in cases where this variable was counterbalanced (Experiments 1 – 2). We used paired and independent samples t-tests for within- and between-group planned comparisons, respectively. Simple effects were used to probe cross-group interactions, which are reported using t-values. An alpha level of. 05 was used across all analyses.

## Bayesian analyses

In some cases, we were interested in whether the predicted pattern of transfer was absent in a group of interest. To assess this, we ran two Bayesian analyses to quantify evidence for the null hypothesis (transfer absent) over the alternative hypothesis (transfer occurred). Our procedures follow closely from those described by Gallistel and colleagues (*Gallistel, 2009*; *Kheifets et al., 2017*) which we highly recommend for interested readers.

To illustrate our approach, we will describe the methodology used for Experiment 1. In the 8-to-4 group of this experiment, the short cue's duration was shifted from 8 to 4 s. Rats were given several sessions to adapt to this cue's new duration before we tested whether this manipulation would cause responding to the long cue to shift leftward.

As a control, we also ran a 'no-change' group that received identical conditions. However, the short cue's duration was left at 8 s throughout. This allowed us to assess whether any variables other than shifting the short cue's duration (e.g. over-exposure to the short cue) could produce an equivalent leftward shift to that seen in the 8-to-4 group.

To assess this, we first computed the marginal likelihood function for the percent change in peak times from training to testing in the no-change group, denoted as:

$$L(\theta|D)$$

For each rat, we estimated the likelihood of obtaining the observed percent change in peak time if it had been drawn from a normal distribution with a mean $\theta$ and a standard deviation equivalent to that seen across rats in the 8-to-4 and no-change groups. Then, we took the log of each rat's likelihood functions, summed across the corresponding values of $\theta$ for each subject, reverted back to a linear scale, and normalized the resulting vector by its sum, such that it integrated to 1. This gave a single, unimodal distribution that gives the estimate of the true posterior distribution from which the data were drawn.

Our null hypothesis was that this distribution had a mean of zero (i.e. no transfer occurred). Therefore, we developed a null prior by following the same procedure described above, yet linearly shifting the data to have mean of zero (i.e. subtracting each value by the sample mean). This distribution is denoted as:

$$\pi(\theta|H_{\sim S})$$

where $H_{\sim S}$ stands for the hypothesis that no shift occurred. The primary question was how much this null distribution overlapped with the group likelihood function, relative to an 'alternative' prior distribution, which was constructed based on the assumption that a shift in responding had indeed occurred during testing.

Our two analyses only differed with respect to how this alternative distribution was constructed. In our 'non-specific shift' analysis, we tested whether any change in peak time occurred, regardless of whether the shift went in the predicted direction (i.e. leftward) or not. Therefore, we used what is often referred to as an 'incremental' alternative prior, denoted as:

$$\pi(\theta|H_S)$$

where $H_S$ stands for the hypothesis that a shift had occurred. In essence, this is a uniform distribution that was centered around the null prior, yet spread out on either side according to the largest

shift seen in the 8-to-4 group and then convolved with the null distribution. Consequently, the distribution assigns a higher prior probability to values that, under the null prior, would be considered unlikely to occur. Therefore, a leftward or rightward peak time shift would cause the alternative distribution to have higher overlap with the group likelihood function, relative to the null prior.

To quantify the degree of evidence in favor of either hypothesis, we first computed a Bayesian posterior between each prior and the group likelihood function:

$$L\big(\theta|D, H_{\sim S/S}\big) = L(\theta|D)\pi(\theta|H)$$

This gives the marginal likelihood of each hypothesis, given the data (i.e. $L(H|D)$). The ratio of the areas for either distribution gives the odds in favor of one hypothesis over another. For simplicity, we computed this as:

$$BF = \frac{L(H_{\sim S}|D)}{L(H_S|D)}$$

The higher the evidence in favor of the null hypothesis, the higher the Bayes factor.

While this approach is effective at detecting evidence for or against a peak time shift during testing, it does not answer our primary question when running the no-change group. Specifically, variables such as overexposure or extinction could very well produce some form of a slight shift during testing. In fact, some evidence does suggest that this could be the case (*Roberts, 1981*).

Importantly, we wanted to test whether these variables would produce an equivalent leftward shift to that seen in the 8-to-4 group. This would falsify our prediction that decreasing the short cue's duration would have anything to do with the a leftward shift in long cue responding seen in the 8-to-4 group.

Therefore, we ran another test that was tailored for this question. Specifically, we constructed an alternative prior using data from the peak-time shifts seen in the 8-to-4 group. The procedures were the same as those used to construct the no-change group likelihood function. If the shifts seen in the no-change group were indistinguishable from those seen in the 8-to-4 group, this alternative distribution should show high overlap with the no-change likelihood function. In contrast, if no (or a slight) change in peak times occurred, the null distribution (i.e. mean of zero) should overlap more closely with the actual data. All other procedures used to quantify the Bayes factor were equivalent to those described above.

The analyses in Experiments 3–4 followed these same methods, yet using data from their respective groups. With respect to heuristics, we refer to Bayes factors > 3 as 'substantial' evidence in favor of the null hypothesis, those >10 as 'strong', and those >100 as 'decisive' (*Gallistel, 2009*). However, we should emphasize that Bayes factors should be interpreted with respect to the continua of evidence that they, in fact, reflect. We do not endorse the assignment of arbitrarily binned cut-offs for what is considered 'real' or not, which we feel is a common problem in typical null-hypothesis testing (i.e. p<0.05).

## Acknowledgements

BJD - Alfred P. Sloan Scholarship, Kwak-Ferguson Fellowship, NIH T32NS007421, and NIH F31NS106737. MSM - NIH R15DA039405

## Additional information

### Funding

| Funder | Grant reference number | Author |
| --- | --- | --- |
| Alfred P. Sloan Foundation | Scholarship | Benjamin J De Corte |
| National Institute of Neurological Disorders and Stroke | NIH T32NS007421 | Benjamin J De Corte |
| National Institute on Drug Abuse | NIH R15DA039405 | Matthew S Matell |

| National Institute of Neurological Disorders and Stroke | NIH F31NS106737 | Benjamin J De Corte |
| Kwak-Ferguson Fellowship | | Benjamin J De Corte |

The funders had no role in study design, data collection and interpretation, or the decision to submit the work for publication.

### Author contributions

Benjamin J De Corte, Conceptualization, Data curation, Software, Formal analysis, Supervision, Validation, Investigation, Visualization, Methodology, Writing—original draft, Writing—review and editing; Rebecca R Della Valle, Data curation, Investigation, Methodology, Writing—review and editing; Matthew S Matell, Conceptualization, Resources, Data curation, Software, Formal analysis, Supervision, Funding acquisition, Validation, Investigation, Methodology, Writing—original draft, Project administration, Writing—review and editing

### Author ORCIDs

Benjamin J De Corte (iD) http://orcid.org/0000-0001-6741-6324
Matthew S Matell (iD) https://orcid.org/0000-0002-5620-8316

### Ethics

Animal experimentation: All procedures accorded with Villanova University's Animal Care and Use Committee guidelines (IACUC, protocol #1880) and the Declaration of Helsinki.

### Decision letter and Author response

Decision letter https://doi.org/10.7554/eLife.38790.034
Author response https://doi.org/10.7554/eLife.38790.035

## Additional files

### Supplementary files

• Source data 1. Compiled data and analyses for all experiments.
DOI: https://doi.org/10.7554/eLife.38790.029

• Transparent reporting form
DOI: https://doi.org/10.7554/eLife.38790.030

### Data availability

Datasets and all functions used for analysis are available as source files associated with the manuscript, both in a compiled (i.e., data/code for all experiments) and figure-specific manner.

The following dataset was generated:

| Author(s) | Year | Dataset title | Dataset URL | Database and Identifier |
| --- | --- | --- | --- | --- |
| De Corte B, Della Valle R, Matell M | 2018 | Data from: Recalibrating timing behavior via expected covariance between temporal cues | http://dx.doi.org/10.5061/dryad.cq2862k | Dryad Digital Repository, 10.5061/dryad.cq2862k |

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
