## [Decision Letter]

Thank you for sending your article entitled "Optimizing future expectations based on the inferred causal structure of the environment" for peer review at *eLife*. Your article is being evaluated by Sabine Kastner as the Senior Editor, a Reviewing Editor, and three reviewers..

Given the list of essential revisions, including new experiments, the editors and reviewers invite you to respond within the next two weeks with an action plan and timetable for the completion of the additional work. We plan to share your responses with the reviewers and then issue a binding recommendation.

Summary:

The authors present a series of experiments testing the hypothesis that expectations for the timing of expected rewards for different cues covary as if determined or at least influenced by an underlying common cause. In support of this proposal, they show that following a change in the timing of reward availability for one cue, behavior for another cue is often changed in the same direction, but to a lesser degree. They dissociate this effect from that predicted by a change in the average duration experienced during the task and show dependence of this effect on correlated changes in duration and context.

Essential revisions:

While the reviewers were overall positive about the significance of the question and the potential impact of the findings, there were significant concerns expressed by several and reinforced on discussion regarding the importance of several control experiments that were not done. It was agreed that the addition of controls for both change and action would be ideal. The former was judged to be particularly important. If one or both of these can be done in a reasonable amount of time, the reviewers felt it would substantially increase support for the common cause hypothesis advanced by the authors. If either or both is not possible, there must be substantial changes to the paper (Abstract, Introduction, Discussion section) to acknowledge these shortcomings or possible caveats. Reviewer 1 also felt that a broader analysis of the behavioral change beyond simply characterizing effect on peak responding is also an essential revision.

*Reviewer #1:*

In the manuscript by de Corte et al., the authors report an elegant series of experiments testing the correlation between temporal expectations induced by different cues on a peak procedure task, when one of those cues undergoes a change in duration.

Overall, the authors nicely demonstrate the shift effect they predict under a 'common cause' hypothesis, in which a shift in temporal expectation induced by changing the trained interval of one cue transfers to the second, previously trained cue, for which the duration has never been changed. They disambiguate this effect from that predicted by a change in the average duration experienced during the task (before the test phase) and show how this duration transfer effect depends on correlated changes in duration for all (shifted) cues and is specific to the context in which the change in duration is experienced.

Some relatively minor amendments should improve the clarity of the manuscript, increase the transparency of the results, and provide a more complete picture of the changes in responding measured in their series of experiments.

My main concern is with the reporting of the change in responding largely in terms of a change in peak responding. While the parametric fit of a Gaussian to the average response curves certainly captures the change in responding as a change in the mean of the fit distribution, it seems in many of the experiments that this obscures some of the finer details of the change in response induced by the experimental manipulation. For instance, start and stop times change differently for some of the experiments, indicating there may be a change as well in the spread of the response curve in some cases? While the start/stop times are reported in the supplemental for most (?) of the experiments, I think some more attention needs to be diverted to this variability in the main Results section.

For instance, the mean+/-SEM of all fit parameters for at least the first two experiments would be very helpful for the reader in interpreting the nature of the change in responding on these tasks.

I also do not recall seeing reference to the start/stop times in the supplemental for the control experiment 2 – since this is a critical control, it seems important to highlight that there is no change in start/stop times as well as in the fit peak times.

There were also some interesting inconsistencies in the experiments that were not discussed but should be addressed. First – the% change in the correlated group of experiment 3 is significant, but much smaller than the other experiments. Is this to be expected under the common-cause hypothesis? Would not more evidence of a shift in durations in this context lead to greater rather than lesser transfer to this cue? The supp methods state there are N=5 per group here but the authors also suggest there are min N=10 elsewhere.

Finally, the discussion links the common-cause hypothesis to latent state inference in rats, but the authors do not cite some influential literature in this domain, especially Wilson et al., 2014 which pre-dates the two hidden state inference citations given. I think this part of the Discussion section would also benefit from linking to latent state inference in behavior more generally as a large body of literature exists in this domain with well-developed and generalizable computational and neural theories (for e.g. Daw, 2005; Doll, 2012).

*Reviewer #2:*

Achievements:

The authors novelly and valuably present four different experiments robustly illustrating that following a change in the timing of reward availability for one cue, behavior for another cue is often changed in the same direction, but to a lesser degree. This is a thorough set of observations for which any theory of timing would have to account. Their findings provide a strong challenge to existing perspectives and are therefore significant for the field and worthy of publication in *eLife*.

Caveats, problems:

The authors should clarify that their results do not demonstrate animals learn or optimally use the causal structure of their environment within the time domain. The authors also do not have controls that show it is really the change in a specific cue-duration relationship that produces the observed effect (although it is very likely), nor can they attribute their findings to a general expectation of timing covariance instead of action-specific mechanisms which involve less representational richness (a more reasonable alternative).

Major rhetoric issues:

The Title and the Abstract both imply that the authors have shown that animals (1) explicitly represent the statistics relating timing of various sources of reward, (2) update a posterior distribution of expected cue-reward times following new evidence from other cues and (3) optimize their timing behavior accordingly. However, the authors do not construct any experiment in which a specific cross-cue timing relationship is knowable, they do not calculate the optimal policy based on such a relationship, and they do not demonstrate that animals follow such an optimal policy. The Title and Abstract should be changed to prevent this misleading interpretation. Maybe "Interval timing shifts suggests animals infer timing correlations" would be a better title. The main text does not similarly mislead.

Alternative to 'common cause' hypothesis 1, and possible control experiment (maybe not necessary):

Animals' timing behavior could be sensitive to the external reward-time structure generally. To control for this possibility, the authors could train sessions that only present a novel cue with a 32s duration in place of sessions that change the 16s duration to 32s for a familiar cue in Experiment 1 and observe the effect on timing following the 8s cue as before. However, the breadth of observations across experiments make the interpretation that the general change in reward timing landscape accounts for all the observed effects, although they make account for some of the effects, less likely.

Alternative to 'common cause' hypothesis 2, and possible control experiment (greatly improve interpretation):

The 'common cause' hypothesis proposes that animals make inferences based on expected positive correlations in timing. However, reinforcement mechanisms that contribute to learning a change in timing for one cue may generally impinge on the expressed timing for other cues, without any specific representation of timing covariance or calculated inference. To prove that some general timing expectation is altered, the authors should show the effect can occur across actions. The 8s cue could be paired with reward after entry into one nosepoke, while the 16s cue is paired with reward after entry into a separate nosepoke. Better yet, the 8s cue could require a lever press, while the 16s cue requires a nosepoke.

*Reviewer #3:*

This manuscript by De Corte et al., argues that rats assume a covariance in different cue-reward delays within an environment and attempt to capitalize on this assumption to infer unobserved intervals when some observable intervals change. They then show that this inference is not behaviorally observable when the assumption of covariance is explicitly broken in the environment. They also show that this effect is context-dependent.

Overall, I like the study and find the authors' hypothesis quite intuitive and interesting. I find the statistics, data analysis, and treatment of prior literature to be satisfactory. The authors also do some obvious controls such as the no change and directionality controls. Nevertheless, there is an important additional control that is imperative before these conclusions can be drawn.

Major Comments

1) The key claim here is that rats are assigning durations to individual cues and inferring that when one cue-reward interval is changed, another is likely to change as well, due to a phylogenetically evolved assumption of covariance. For this to be shown, it is required to demonstrate that the change in the test intervals happens solely due to the change in the observed cue-reward interval change for the other cue. However, the results observed could be explained even if the animals ignore the causal structure.

For instance, an important control experiment for Figure 1A is this: Train Cue1-8 s and Cue2-16 s. For the Change phase, add Cue3-4 s instead of Cue1-4 s. Show that in this case, Test for Cue2 shows no change. If this experiment showed a change in the test for Cue2, then that means that the effect is not due to an inferred "causal structure" but instead because the relevant intervals to be produced generally decreased. Since this explanation could work for all experiments presented, a similar control needs to be done for each experiment to rule out an explanation of a general adaptation to change in timed intervals.

2) I find it interesting that in the only experiment in which the assumption of covariation was explicitly confirmed to the animals (Figure 4A), the% change in peak time seems to be lower in magnitude than the experiments in which there is no explicit information regarding whether the covariance assumption is valid (Figure 1). If the "common cause" hypothesis held, would one not expect that if the animals had explicit confirmation of a "common cause" like in Figure 4A, their behavior would show bigger changes, especially considering that this is argued to be an "optimal" inference?

---

## [Author Response]

[Editors' note: the authors’ plan for revisions was approved and the authors made a formal revised submission.]

Reviewer #1:[…] My main concern is with the reporting of the change in responding largely in terms of a change in peak responding. While the parametric fit of a Gaussian to the average response curves certainly captures the change in responding as a change in the mean of the fit distribution, it seems in many of the experiments that this obscures some of the finer details of the change in response induced by the experimental manipulation. For instance, start and stop times change differently for some of the experiments, indicating there may be a change as well in the spread of the response curve in some cases? While the start/stop times are reported in the supplemental for most (?) of the experiments, I think some more attention needs to be diverted to this variability in the main Results section.For instance, the mean+/-SEM of all fit parameters for at least the first two experiments would be very helpful for the reader in interpreting the nature of the change in responding on these tasks.I also do not recall seeing reference to the start/stop times in the supplemental for the control experiment 2 – since this is a critical control, it seems important to highlight that there is no change in start/stop times as well as in the fit peak times.

We appreciate your suggestions to provide more detail on the behavioral results. We now provide the mean change +/- SEM for all reported timing indices across experiments. There were indeed changes in the spread of unchanged-cue responding. However, these effects were largely expected due to ‘scalar timing’, in which the spread of responding scales in direct proportion to the peak time (for discussion see Roberts, 1981; Buhusi and Meck, 2005). For example, if the peak time were to double (for any reason), the spread would also be expected to double, due to this timing property (see text for further description; Results section). To address this, we now report the coefficient of variation (CV = spread/peak time) data for all experiments. This is a common way to characterize spread changes, while accounting for scalar timing, in this task. Specifically, if spread changes were due to scalar timing, this normalized-spread value should remain constant, relative to training. This result was seen in virtually all experiments. There are a few minor CV effects in some cases, but these effects were neither systematic nor restricted to the critical cue.

Your note on differential effects on start/stop times is also important. Some manipulations can indeed selectively impact start or stop times (e.g., dopamine manipulations often affect start times but not stop times; see Balci, 2014). These results can have substantial theoretical relevance from a decision-making standpoint, so we have brought more attention to them in the main Results section for all experiments as well. As we noted in the results of Experiment 1, our single-trial data were limited and noisy due to extinction. Nonetheless, the overall pattern of start/stop time changes across experiments does not suggest a preferential effect of the transfer on either measure.

Finally, the bar graph detailing start/stop time changes in the control group of Experiment 1 was mistakenly added to Figure 1—figure supplement 2 of the initial version (see top left corner). This figure was supposed to present data from groups not included in Experiment 1. These data are now correctly included in Figure 1—figure supplement 1, and thank you for pointing this out.

There were also some interesting inconsistencies in the experiments that were not discussed but should be addressed. First – the% change in the correlated group of experiment 3 is significant, but much smaller than the other experiments. Is this to be expected under the common-cause hypothesis? Would not more evidence of a shift in durations in this context lead to greater rather than lesser transfer to this cue?

Yes, while the transfer was consistent with our hypothesis, we also felt that the magnitude of the transfer was surprisingly small in the correlated group. We expected it to exceed or, at the very least, match the 16-to-32 group of Experiment 1. However, as we describe in the revised manuscript, we believe within-modality categorization influenced our results in this case. Specifically, in the correlated group of Experiment 3 (now Experiment 4a), the changed cues (short and long) were both auditory stimuli and the unchanged cue (medium) was a visual stimulus. We suspected that this could have caused rats to begin categorizing the changed cues as belonging to the auditory modality alone, thereby weakening the transfer to the critical (visual) cue. The other experiments in the initial manuscript only presented one cue from each modality, which would have prevented modality-specific categorization.

To assess this, we ran a follow up study in which the short and long (i.e., changed) cues were visual and auditory, respectively, preventing modality categorization. As predicted, transfer to the medium cue (auditory in this case) was nearly equivalent to that seen in Experiment 1. We have added this follow up as Experiment 4b and the initial experiment is now listed as Experiment 4a.

The supp methods state there are N=5 per group here but the authors also suggest there are min N=10 elsewhere.

We have clarified this in the main results (figure captions) and Materials and methods sections for each experiment. Specifically, all experiments had a minimum of 10 rats, yet some included sub-groups with minimum *ns* of 5 (this is also now stated more clearly in the Supplemental Methods).

Finally, the discussion links the common-cause hypothesis to latent state inference in rats, but the authors do not cite some influential literature in this domain, especially Wilson et al., 2014 which pre-dates the two hidden state inference citations given. I think this part of the Discussion section would also benefit from linking to latent state inference in behavior more generally as a large body of literature exists in this domain with well-developed and generalizable computational and neural theories (for e.g. Daw, 2005; Doll, 2012).

Thank you for recommending this literature, which is now cited/discussed in-text. We have revised the subsection ‘Causal inference in rats?’ to frame the results in terms of model-free and model-based mechanisms (subsection is now titled ‘Model free vs. model-based mechanisms’). We enjoyed conceptualizing the data in this context as it provides a broader perspective for our results. It also helped us clarify the current boundaries of the common cause hypothesis, as detailed further in response to reviewer 2.

Reviewer #2:[…] Caveats, problems:The authors should clarify that their results do not demonstrate animals learn or optimally use the causal structure of their environment within the time domain. The authors also do not have controls that show it is really the change in a specific cue-duration relationship that produces the observed effect (although it is very likely), nor can they attribute their findings to a general expectation of timing covariance instead of action-specific mechanisms which involve less representational richness (a more reasonable alternative).Major rhetoric issues:The Title and the Abstract both imply that the authors have shown that animals (1) explicitly represent the statistics relating timing of various sources of reward, (2) update a posterior distribution of expected cue-reward times following new evidence from other cues and (3) optimize their timing behavior accordingly. However, the authors do not construct any experiment in which a specific cross-cue timing relationship is knowable, they do not calculate the optimal policy based on such a relationship, and they do not demonstrate that animals follow such an optimal policy. The Title and Abstract should be changed to prevent this misleading interpretation. Maybe "Interval timing shifts suggests animals infer timing correlations" would be a better title. The main text does not similarly mislead.

These are certainly fair points. In hindsight, the initial title and some of the text were certainly overly ambitious. We have made the following revisions to address these concerns. First, as noted above, we have toned down the title as follows:

Recalibrating timing behavior via expected covariance between temporal cues.

This removes the ‘optimality’ reference and our use of the term ‘timing behavior’ is intended to avert the implication that temporal expectations, per se, are being changed across phases (over other mechanisms such as reinforcement-based effects you mention, see below).

Second, we removed optimality/inference references in the Abstract and main text.

Third, we have added a paragraph to the main Discussion section emphasizing that evaluating the (potential) optimality of the transfer is a critical future direction as follows (subsection “Future directions and beyond timing”):

“Finally, an important question is whether the transfer follows an optimal policy. As noted above, the common cause hypothesis can primarily predict whether transfer will occur and in what direction. However, we also propose that the transfer serves an adaptive function. If this were the case, one would expect that the degree of transfer would follow a systematic optimization principle, based on the expected covariance between the cues. To address this, we recommend explicitly training the animals on what degree of temporal covariance to expect between cues by including phases in which all cues are reinforced at new times and the degree of the shifts follows a programmed covariance relationship. If only a single cue were shifted during a final change phase, one could evaluate whether responses to the remaining cue(s) follow the experimentally chosen covariance policy.”

Alternative to 'common cause' hypothesis 1, and possible control experiment (maybe not necessary):Animals' timing behavior could be sensitive to the external reward-time structure generally. To control for this possibility, the authors could train sessions that only present a novel cue with a 32s duration in place of sessions that change the 16s duration to 32s for a familiar cue in Experiment 1 and observe the effect on timing following the 8s cue as before. However, the breadth of observations across experiments make the interpretation that the general change in reward timing landscape accounts for all the observed effects, although they make account for some of the effects, less likely.

We agree fully and, as mentioned in our revision plan, had the same concern. We do believe that some of our initial results, particularly from Experiments 1-2, were sufficient to weaken this explanation (described in the introduction to Experiment 3). Experiments 4a and 4b (correlated vs. uncorrelated groups) also pose a problem, as the reward structure was altered in the uncorrelated groups in a comparable way to Experiment 1 and no reliable shift was seen in either case (not explicitly addressed in-text, for brevity). However, we agree that a more direct test was needed. Therefore, we have added the following results as Experiment 3 in the manuscript. The design is similar to the one you recommend. We initially trained two groups to associate distinct cues with either an 8- or 16-second delay to reward. Once trained, we introduced a second phase, in which the 16-second cue was not presented, but a novel cue was introduced. In the ‘no-change group’, the novel cue was associated with a 4-second duration, and the initial 8-second cue was presented, with its duration unchanged. Conversely, in the ‘change group’, the 8-second cue’s duration was shifted to 4-seconds, and the novel cue’s duration was set at 8-seconds. Thus, both groups had the same reward structure (cues, durations, etc.). However, only the change-group experienced a cue-duration change. Therefore, any group differences in transfer to the 16-second cue at test would have to be caused by a specific cue-duration change.

As predicted, the transfer to the 16-second cue was more than 3 times larger in the change group than the no-change group, confirming that cue-duration changes strongly regulate the effect. Admittedly, there was some evidence of transfer, albeit more mild, in the no-change group (although only significant for stop times). Thus, while there may be some contributions from overall changes in reward delays, the results indicate that specific cue-duration changes far exceed the reward-structure effects, which was the critical question in this case.

Finally, although it was not explicitly requested, we had hoped to have the reverse version of this change-control experiment (i.e., shifting to a longer duration, rather than a shorter one) completed by the time of resubmitting as well. Unfortunately, a variety of errors/equipment problems (e.g., feeder jams, power outages) occurred during the process that resulted in disrupted behavior and made the experiment invalid. We would certainly be willing to rerun and add these data if the review group feels they are critical, but doing so would further delay resubmission.

Alternative to 'common cause' hypothesis 2, and possible control experiment (greatly improve interpretation):The 'common cause' hypothesis proposes that animals make inferences based on expected positive correlations in timing. However, reinforcement mechanisms that contribute to learning a change in timing for one cue may generally impinge on the expressed timing for other cues, without any specific representation of timing covariance or calculated inference. To prove that some general timing expectation is altered, the authors should show the effect can occur across actions. The 8s cue could be paired with reward after entry into one nosepoke, while the 16s cue is paired with reward after entry into a separate nosepoke. Better yet, the 8s cue could require a lever press, while the 16s cue requires a nosepoke.

This is a very interesting idea that we hadn’t considered. This ties in well with the model-free vs. model-based discussion suggested by reviewer 1, so we have incorporated your critique as follows (subsection “Model-free vs. Model-based mechanisms?”):

“Particularly important consideration is how simple extinction and reinforcement mechanisms could produce the transfer.

Specifically, in our tasks, changing a cue’s duration entailed extinguishing responses emitted at the old reinforced time and reinforcing responses emitted at the new time. If these response-level extinction and reinforcement processes carried over to unchanged cue trials, responding would shift in the direction of the changed cue’s new duration. To assess this, we recommend using a design in which each cue is associated with a different action-outcome contingency (e.g., Cue 1: 8 seconds/nosepoke; Cue 2: 16 seconds/lever press). During the change phase, model-free processes would operate on different actions. Therefore, under this account, one would not expect the transfer to occur.”

Reviewer #3:[…] Major Comments1) The key claim here is that rats are assigning durations to individual cues and inferring that when one cue-reward interval is changed, another is likely to change as well, due to a phylogenetically evolved assumption of covariance. For this to be shown, it is required to demonstrate that the change in the test intervals happens solely due to the change in the observed cue-reward interval change for the other cue. However, the results observed could be explained even if the animals ignore the causal structure.For instance, an important control experiment for Figure 1A is this: Train Cue1-8 s and Cue2-16 s. For the Change phase, add Cue3-4 s instead of Cue1-4 s. Show that in this case, Test for Cue2 shows no change. If this experiment showed a change in the test for Cue2, then that means that the effect is not due to an inferred "causal structure" but instead because the relevant intervals to be produced generally decreased. Since this explanation could work for all experiments presented, a similar control needs to be done for each experiment to rule out an explanation of a general adaptation to change in timed intervals.

This is a very good point also raised by reviewer 2. We now include a new experiment, which confirms that specific cue-duration changes strongly drive the transfer (see Experiment 3 as well as response to reviewer 2).

2) I find it interesting that in the only experiment in which the assumption of covariation was explicitly confirmed to the animals (Figure 4A), the% change in peak time seems to be lower in magnitude than the experiments in which there is no explicit information regarding whether the covariance assumption is valid (Figure 1). If the "common cause" hypothesis held, would one not expect that if the animals had explicit confirmation of a "common cause" like in Figure 4A, their behavior would show bigger changes, especially considering that this is argued to be an "optimal" inference?

Also a very good point, which was raised by the other reviewers. We believe the results of Experiment 4b help to address this caveat (see responses and main text). Those results show shifts that are roughly the same magnitude as seen in Experiment 1. We can only speculate as to why the transfer was not magnified, relative to Experiment 1. The obvious possibility is that rats weigh expected covariance equally to confirmed covariance. A related possibility is that there is a ceiling effect on the transfer. Our suspicion is that the transfer reflects a weighted average of the unchanged cue’s initial duration and what its duration would be if it had shifted by the same factor as that observed for the changed cue(s). They may simply adjust the weight given to the initial duration to ensure that their estimate does not deviate too far from this initial value. Unfortunately, we don’t have data necessary to address this issue (see notes on optimality below). Nevertheless, we don’t feel that this is absolutely critical to address before moving forward with the manuscript.